# Hidden vulnerability of US Atlantic coast to sea-level rise due to vertical land motion

Leonard O. Ohenhen [1,2] ✉, Manoochehr Shirzaei [1,2], Chandrakanta Ojha[3] & Matthew L. Kirwan [4]

The vulnerability of coastal environments to sea-level rise varies spatially, particularly due to local land subsidence. However, high-resolution observations and models of coastal subsidence are scarce, hindering an accurate vulnerability assessment. We use satellite data from 2007 to 2020 to create high-resolution map of subsidence rate at mm-level accuracy for different land covers along the ~3,500 km long US Atlantic coast. Here, we show that subsidence rate exceeding 3 mm per year affects most coastal areas, including wetlands, forests, agricultural areas, and developed regions. Coastal marshes represent the dominant land cover type along the US Atlantic coast and are particularly vulnerable to subsidence. We estimate that 58 to 100% of coastal marshes are losing elevation relative to sea level and show that previous studies substantially underestimate marsh vulnerability by not fully accounting for subsidence.

Coastal zones—the low-elevation (<10 meters above sea level) zone at the land-water interface – provide essential habitat, ecosystem, and environmental functions. Landward, coastal areas host one-third of the world's population, and 15 out of 20 present-day megacities are located within low-elevation coastal zones[1]. Toward the sea, most coastlines are sheltered by coastal wetland ecosystems, providing invaluable physical, chemical, biological, and socioeconomic benefits such as food production, water filtration, preservation of biodiversity, shoreline protection, storm buffering, sediment retention, nutrient cycling, and carbon sequestration[2–8]. Due to their utility and dynamism, coastal zones are highly vulnerable and sensitive to hazards related to changing environmental and climatic conditions[9–13].

Coastal areas are particularly vulnerable to the effects of global climate change[14], which has caused sea level rise (SLR) to accelerate from ~1.7 to ~3.35 mm per year in the last century[15], with a predicted rise of 1 m or more by 2100 (ref. 16). Driven by SLR, coastal zones are facing serious threats from coastal flooding, erosion, storms, and saltwater incursion into estuaries and coastal aquifers[10–13,17]. These hazards are exacerbated by regional influences on relative SLR, such as ocean currents, coast morphology, and local rates of vertical land motion (VLM) (i.e., subsidence or uplift, including changes in surface elevation due to deposition or erosion)[18]. Subsidence associated with local fluid extraction, sediment compaction, and aquifer-system compaction significantly influence relative sea levels. Subsidence-influenced relative sea levels may exacerbate flood risks, promote the salinization of soil and water supplies, corrosion and weakening of infrastructures, and the devaluation, decreased functionality, and loss of wetlands[9,19–24].

The US Atlantic coast is the most populous coast in the US, hosting more than a third of the US population. Portions of this coast are typically recognized as a hotspot of SLR[25], primarily due to widespread land subsidence related to the compaction of young sediments, groundwater extraction[13,14,18], and glacial isostatic adjustment (GIA)[26]. This ~3,500 km long coast hosts a diverse coastal ecosystem, including 36 million hectares of wetlands worth ~US$360 billion[27–29]. Relative SLR has caused an increase in coastal flooding in the region, particularly during the last decade[30] (Fig. 1). Increased flooding has resulted in the frequent disruption of economic activities in several cities on the US Atlantic coasts and the prolonged inundation of wetland ecosystems[31–35]. The loss of wetlands may result in an increase in the susceptibility, risk levels, and vulnerability of the entire coast to coastal hazards[36].

Because of the spatially variable coastal ecosystem characteristics and dynamic nature of coastal risk, current observations and models of

¹Department of Geosciences, Virginia Tech, Blacksburg, VA, USA. ²Virginia Tech National Security Institute, Blacksburg, VA, USA. ³Department of Earth and Environmental Science, IISER Mohali, Punjab, India. ⁴Virginia Institute of Marine Science, College of William and Mary, Gloucester Point, Virginia, USA. ✉e-mail: ohleonard@vt.edu

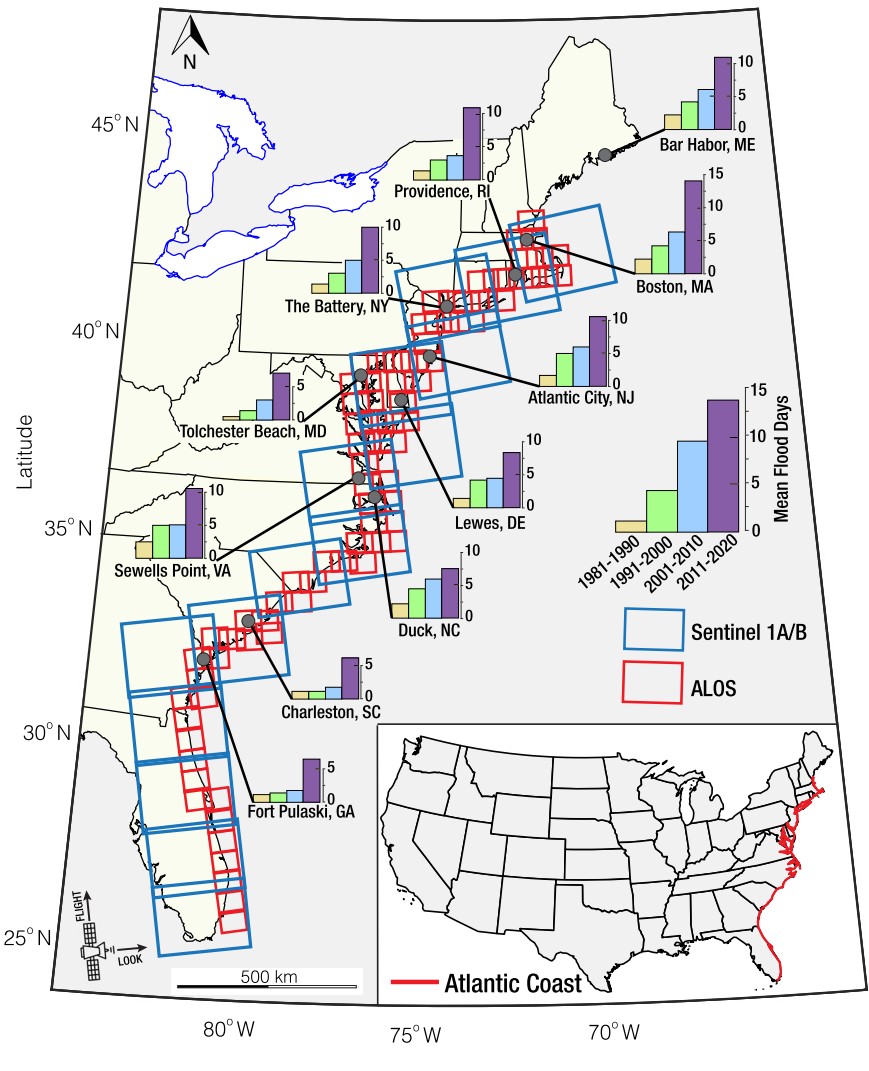

**Fig. 1 | Study area and Datasets.** The study area is the US Atlantic coast (New Hampshire to Florida). The interferometric datasets include acquisition from Sentinel-1A/B C-band satellites (blue rectangles) spanning 2015–2020 obtained in ascending orbits and advanced land observing satellite (ALOS) L-band (red rectangles) spanning 2007–2011 and acquisitions obtained in and ascending orbits. The path and frame parameters of the SAR datasets are summarized in Supplementary table 1. Frequency of flooding along the US Atlantic coast are shown as bar charts. This bar chart shows the average number of flood days per decade at 11 tide gauge stations along US Atlantic coasts[30]. Each small bar graph compares each decade: 1981–1990 in yellow, 1991–2000 in green, 2001–2010 in blue, and 2011–2020 in purple. Note: The tide gauge station names are displayed with each bar chart. State Codes: ME Maine, MA Massachusetts, NY New York, NJ New Jersey, MD Maryland, DE Delaware, VA Virginia, NC North Carolina, SC South Carolina, and GA Georgia. U.S. National, state, and great lakes boundaries are based on public domain vector data by World DataBank (https://data.worldbank.org/) generated in MATLAB.

coastal vulnerabilities are inadequate due to sparse and often point-wise measurements of VLM[4,9,37,38]. Due to their vast extent and difficult terrain, coastal wetlands monitoring is a challenging endeavor that restricts ground-based operations[39]. While field observations may provide accurate in-situ data for wetland monitoring, these techniques are impractical for large-scale and frequent wetland monitoring[40]. Moreover, comparisons between wetland accretion data obtained from ground-based observations and relative SLR do not fully account for both shallow and deep subsidence[38], so existing coastal wetland vulnerability assessments likely underestimate the vulnerability of coastal ecosystems[41–45]. Interferometric synthetic aperture radar (InSAR) overcomes these challenges providing unprecedented spatio-temporal resolution on elevation change and has proved useful for monitoring various land cover types and analyzing dynamic wetland changes[39,40,46–50]. InSAR measurements provide cost-effective and up-to-date wetland data, which offers improved accuracy and increased spatial density and extent of VLM measurements[19,20,51].

Here, we present VLM rate map for the US Atlantic coast at ~50-m resolution and mm-level precision using a combination of Sentinel-1 and Advanced Land Observing Satellite (ALOS) satellites with observations at global navigation satellite system (GNSS) stations. This spatially semi-continuous map shows broad-scale patterns of subsidence exceeding 3 mm per year across the US Atlantic coast with some localized zones of uplift. We explore the exposure of different coastal ecosystems in the region using the produced VLM rates with wetlands, forests, agricultural areas, and developed areas having the most significant exposure to subsidence. Using these VLM rates, we provide vulnerability estimates of coastal wetlands for the US Atlantic coast.

## Results
### Spatially semi-continuous vertical land motion for the US Atlantic coast
We combined several synthetic aperture radar (SAR) datasets from Sentinel-1A/B (2015–2020) and ALOS (2007–2011) satellites with 173

GNSS observations (2007–2020) to characterize land deformation within ~100 km inland of the US Atlantic coast. The SAR dataset included several frames collected in ascending orbit geometry (Fig. 1 and Supplementary Table 1). The combined datasets were processed using a multitemporal wavelet-based InSAR (WabInSAR) algorithm[52,53] to generate three-dimensional (3D) line-of-sight (LOS) velocity (Supplementary Fig. 1, see methods). A unified weighted least-squares joint optimization model was then applied to combine the LOS velocities with the horizontal and vertical displacement velocities of the GNSS observations to determine the 3D deformation field at each of the ~38 million elite pixels. The model validation was implemented via an analysis of the standard deviation (SD) associated with each InSAR pixel (i.e., precision) and a comparison of the InSAR VLM with independent GNSS data as ground truth (i.e., accuracy) (see methods and supplementary Figs. 2 and 3).

The horizontal (east and north) velocities along the US Atlantic coast are shown in Supplementary Fig. 2a, c. The horizontal rate shows the relative motion of the North American plate in the northwest direction, consistent with earlier works[54]. Figure 2a shows the VLM rates with positive values indicating uplift and negative values indicating subsidence. The VLM map shows mostly subsidence, with some

localized uplift in the region. From the ~38 million pixels map of VLM, 90% of the pixels show subsidence (Supplementary Fig. 4), highlighting the broad-scale spatial pattern of subsidence across the US Atlantic coast. The subsidence rates across the east coast are spatially variable, with rates exceeding 3 mm per year in most cities (Fig. 2c). The major cities undergoing subsidence in the region are Boston, MA; New York, NY; Atlantic city, NJ; Lewes, DE; Norfolk, VA; and Charleston, SC. In Charleston, Brunswick, and the Chesapeake Bay, subsidence exceeds 5 mm per year (Fig. 2c). High subsidence rates in the Chesapeake Bay are primarily observed in areas surrounding Delaware Bay and the Peninsula, with some localized uplift. This observed uplift is consistent with the current groundwater recharge rate in the region, manifested in rising hydraulic head levels[55] (Supplementary Fig. 5).

Sea level projections rarely accurately account for local land subsidence, contributing to uncertainties in assessing sea level change. For instance, the rates of VLM in the IPCC Sixth Assessment Report of global sea level change projections[56,57] use a constant long-term background rate of change estimated from historical tide gauge trends[57,58]. Estimates of VLM from tide gauges do not account for subsidence in shallow strata and are likely to underestimate VLM or represent only minimum values[57]. To assess uncertainties in the

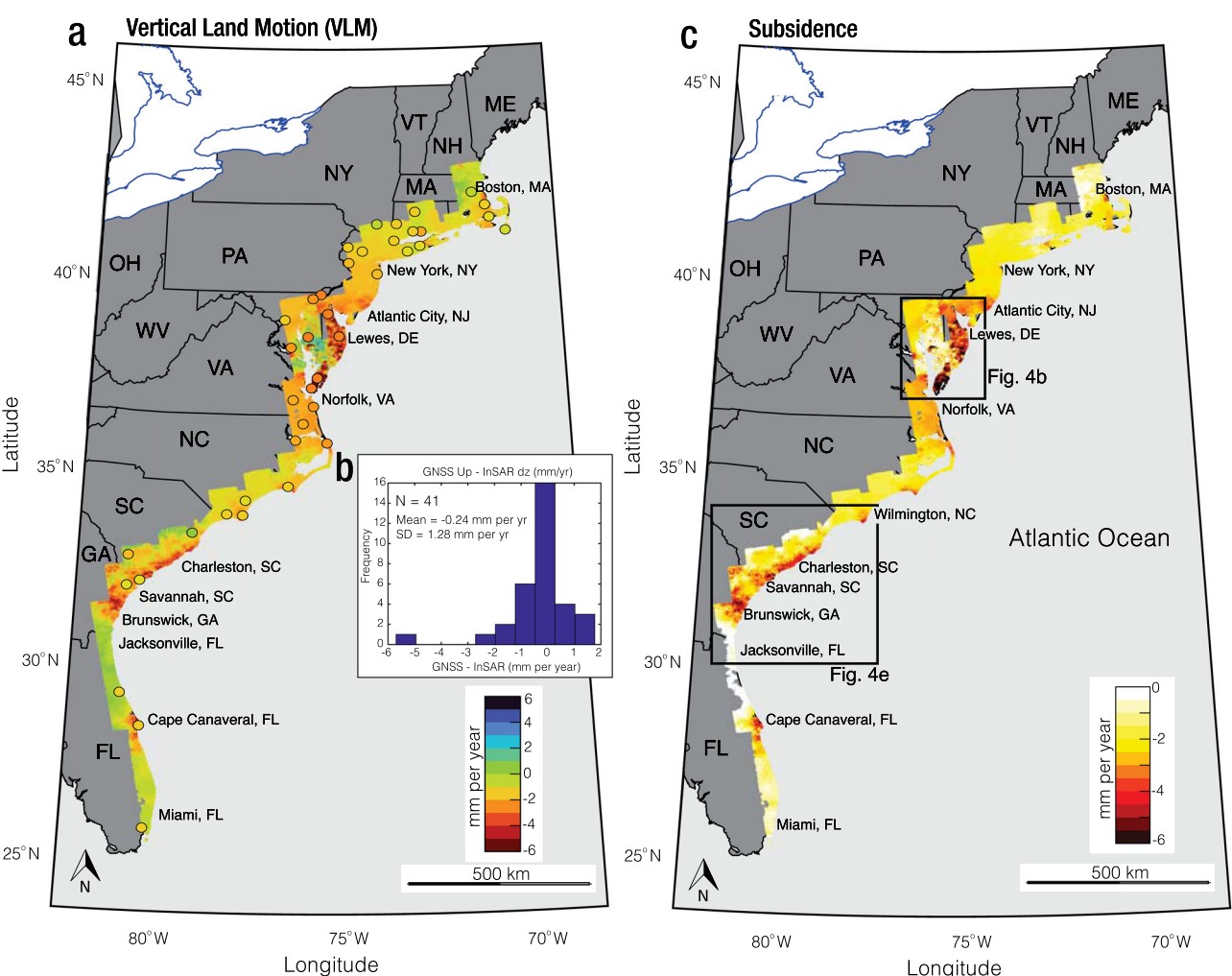

**Fig. 2 | Vertical land motion (VLM) across the US Atlantic coast. a** Estimated VLM rate. The circles show the location of GNSS validation observations color-coded with their respective vertical velocities. **b** Histogram comparing GNSS vertical rates with estimated VLM rates. The standard deviation (SD) of the difference between the two datasets is 1.3 mm per year. **c** Land subsidence (representing negative VLM) across the US Atlantic Coast. The black rectangles indicate the extent of study areas for Chesapeake Bay area and Georgia, South Carolina, and North Carolina (GA-SC-

NC) area shown in Fig. 4. State Codes: ME Maine, NH New Hampshire, VT Vermont, MA Massachusetts, RI Rhode Island, NY New York, PA Pennsylvania, NJ New Jersey, WV West Virginia, OH Ohio, DE Delaware, VA Virginia, NC North Carolina, SC South Carolina, GA Georgia, and FL Florida. National, state, and great lakes boundaries in **a**, **c** are based on public domain vector data by World DataBank (https://data.worldbank.org/) generated in MATLAB.

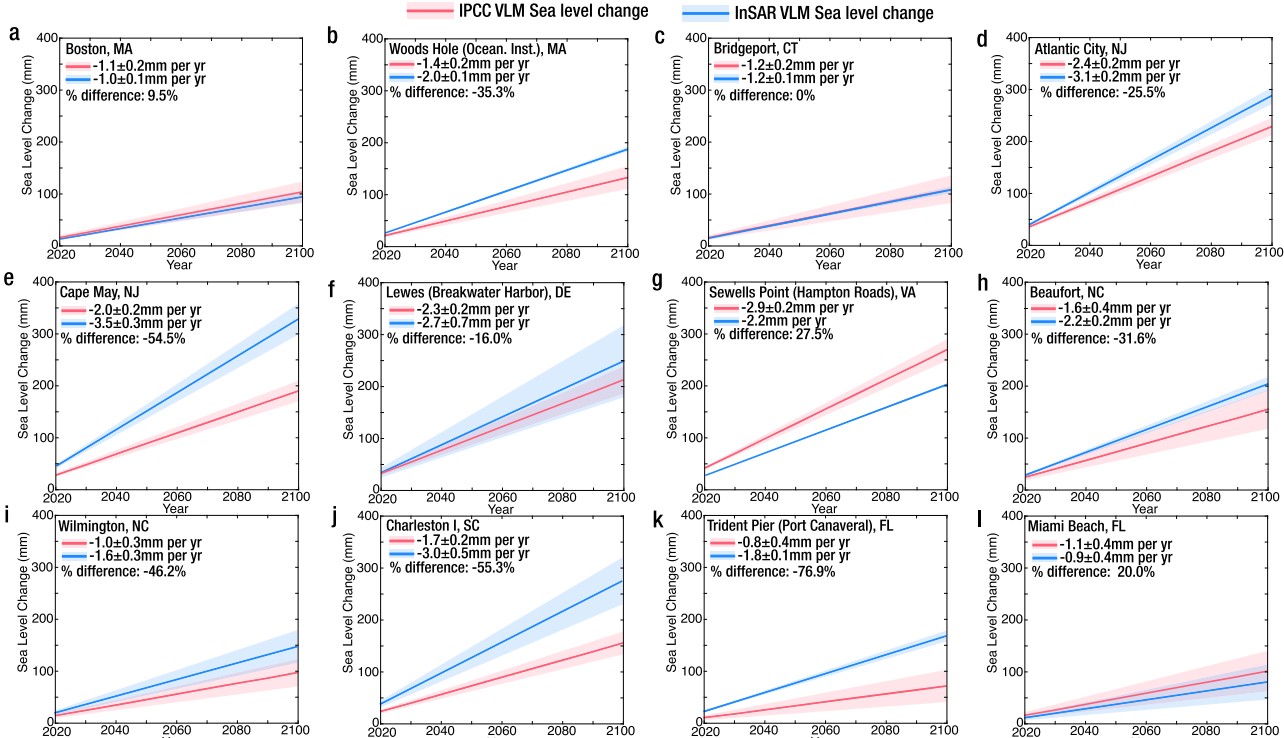

**Fig. 3 | Sea-level change due to vertical land motion (VLM).** Comparison of sea-level change (millimeters) due to vertical land motion from IPCC Sixth Assessment Report[56,57] with sea-level change (millimeters) using VLM estimate from this study at 12 tide gauge stations. **a** Boston, Massachusetts (MA), **b** Woods Hole, MA, **c** Bridgeport, Connecticut (CT), **d** Atlantic City, New Jersey (NJ), **e** Cape May, NJ, **f** Lewes, Delaware (DE), **g** Sewells Point, Virginia (VA), **h** Beaufort, North Carolina (NC), **i** Wilmington, North Carolina (NC), **j** Charleston I, South Carolina (SC), **k** Trident Pier, Florida (FL), **l** Miami Beach, FL. The solid red line shows the IPCC current sea level projection data, we compared a linear projection of our VLM rates from 2020 to 2100 with the linear projection of VLM used in the IPCC over the same period at 12 tide gauge stations on the US Atlantic coast (Fig. 3 and Supplementary Table 2). To compare the IPCC's VLM rates with the contemporaneous rate of our InSAR VLM, we averaged the VLM rates of InSAR pixels within a 200 m radius of the tide gauges to reduce localized high VLM rates, we used the SD of each InSAR VLM to estimate the error ranges (see methods). The results show that the IPCC sea level change due to VLM was underestimated (considering the % difference) at 7 tide gauge stations (Woods Hole, Atlantic City, Cape May, Beaufort, Wilmington, Charleston I, and Trident Pier; Fig. 3 and Supplementary Table 2) and overestimated (considering the % difference) at 1 tide gauge station (Sewells Point; Fig. 3 and Supplementary Table 2), with 4 tide gauge stations within a 20% difference error range (Boston, Bridgeport, Lewes, and Miami Beach; Fig. 3 and Supplementary Table 2). At Cape May, Charleston I, and Trident Pier, the contrasts between the IPCC projected VLM rates and the projected InSAR VLM rates are significant (greater than 1 mm per year), which highlights significant uncertainties in the rates of local and regional projected relative sea level change.

## Coastal land cover exposure to subsidence

Using the US Geological Survey (USGS) 2019 National Land Cover (NLC) map[29], we estimated the exposure of different coastal systems along the US Atlantic coast to subsidence. To this end, we interpolated VLM rates over NLC pixels, and exposure to subsidence is defined based on the percentage of InSAR pixels in each land cover type. The NLC is a 30-meter grid map, which provides spatial characteristics of

VLM sea-level change is 50th percentile and the shaded ranges show the 5th–95th percentile. The solid blue line shows the InSAR VLM sea-level change from this study, and the shaded ranges are 2 standard deviations. Note: the negative rates of VLM (or subsidence) result in a positive sea level change. The blue shaded region is narrow at some tide gauges due to small standard deviation values (e.g., Woods Hole, Bridgeport, and Sewells Point). The percent (%) difference is calculated as: $\frac{IPCC\ VLM - InSAR\ VLM}{(IPCC\ VLM + InSAR\ VLM/2)} \times 100$. A negative % difference indicates underestimation and a positive difference indicates overestimation, with a ±20% error buffer.

the land surface, such as area of developed land, barren land, forests, shrub/scrub, grassland, pasture and hay, cultivated crops, and wetlands (Fig. 4a, d). The NLC and their corresponding percentage of land cover on the US Atlantic coast are wetlands, forests, cultivated crops, and developed regions, which account for 44.5%, 26.3%, 16.8%, and 8.1%, respectively (Table 1). The average rate of VLM for the major NLC classifications ranges from −1.3 mm per year to −1.7 mm per year (Table 1). Agricultural lands (and barren lands) along the US Atlantic coast show some of the highest rates of subsidence in all measured NLC types, with a mean subsidence rate and SD of 1.7 ± 1.3 mm per year (see methods section for a description of the SD associated with the NLC). The maximum rate of sinking for agricultural areas/cultivated crops exceeds 11.8 mm per year, the maximum subsidence rate observed for any NLC type in this study. The areas the NLC noted as barren lands represent the mostly coastal rock/sand/clay strip bordering the Atlantic Ocean. Subsidence on barren lands may be the result of retreating barren lands due to erosion in response to rising sea levels.

The NLC types exposed to subsidence are highlighted in Fig. 4 and Table 1 for two regions with exceptionally rapid subsidence rates. The first region is the Chesapeake Bay area, the largest estuary in the US, where subsidence has been extensively documented in existing literature[59–62]. For the Chesapeake Bay area, subsidence occurs mainly in forests, wetlands, and cultivated crops (Fig. 4 and Table 1). The subsidence rate for the major NLC types are 2.6 ± 1.4 mm per year, 2.5 ± 1.4 mm per year, and 2.1 ± 0.8 mm per year for wetlands, agricultural lands, and forests, respectively (Table 1). In the Chesapeake Bay's Delmarva area, agrarian lands (cultivated crops and Pasture/Hay)

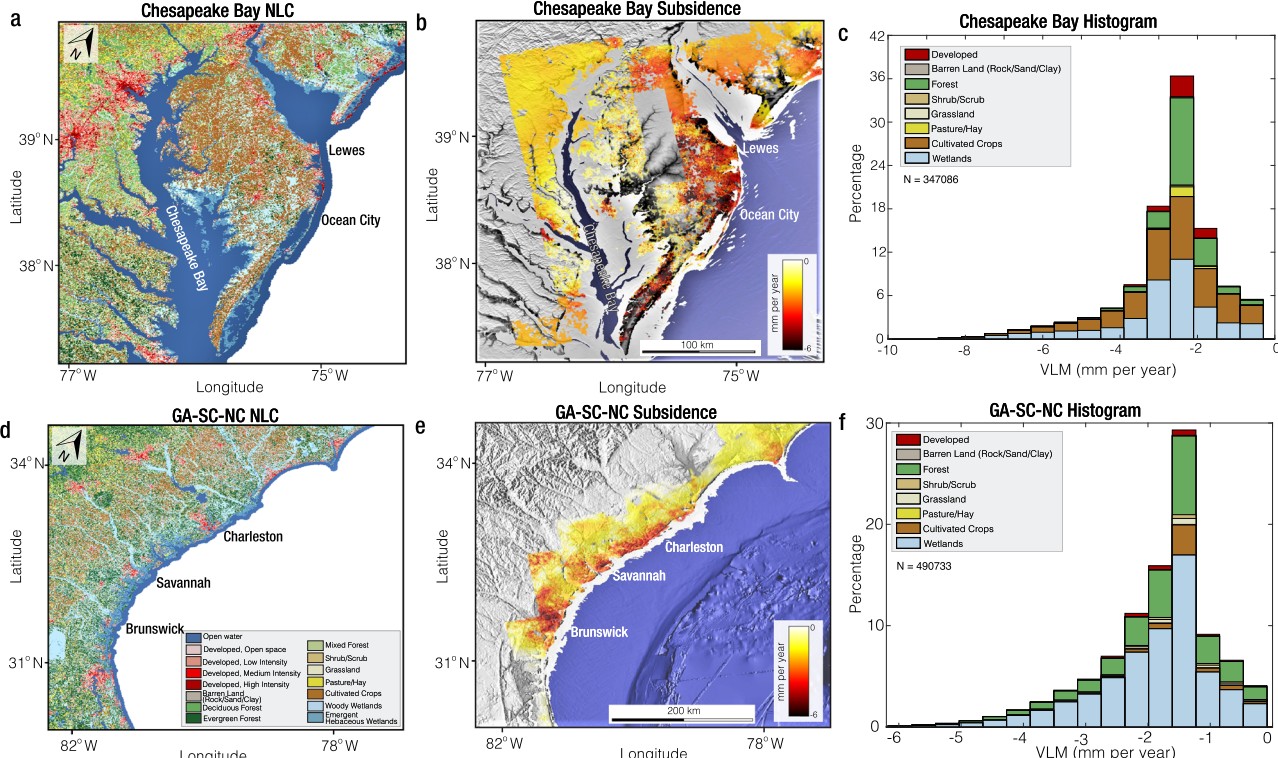

**Fig. 4 | Examples of regions with significant exposure to land subsidence.**
**a** National land cover (NLC) map for the Chesapeake Bay area[29]. **b** Land Subsidence across the Chesapeake Bay area. Background is the global multi-resolution topography (GMRT) made with GeoMapApp (www.geomapapp.org)[100]. **c** Frequency distribution of subsidence for different NLC in the Chesapeake Bay area. The percentage of exposure to subsidence and the average subsidence rate for the different NLC are shown in Table 1. N is the number of pixels and VLM is vertical land motion. **d** NLC map for the Georgia, South Carolina, and North Carolina (GA-SC-NC) area[29]. **e** Land Subsidence across the GA-SC-NC coast. Background is GMRT made with GeoMapApp (www.geomapapp.org)[100]. **f** Frequency distribution of subsidence for different NLC in the GA-SC-NC area. The percentage of exposure to subsidence and the average subsidence rate for the different NLC are shown in Table 1. N is the number of pixels and VLM is vertical land motion.

### Table 1 | National land cover (NLC) exposure to land subsidence

| NLC | US Atlantic Coast | | Chesapeake Bay | | GA-SC-NC | |
|---|---|---|---|---|---|---|
| | **% Exposure** | **VLM (mm per year)** | **% Exposure** | **VLM (mm per year)** | **% Exposure** | **VLM (mm per year)** |
| Developed | 8.1 | −1.3 ± 0.8 | 5.4 | −2.1 ± 0.7 | 2.0 | −1.6 ± 0.8 |
| Barren Land | 0.5 | −1.7 ± 1.1 | 0.4 | −2.8 ± 1.5 | 0.3 | −2.1 ± 1.2 |
| Forest | 26.3 | −1.4 ± 0.9 | 20.2 | −2.1 ± 0.8 | 27.1 | −1.6 ± 0.9 |
| Shrub/Scrub | 0.7 | −1.2 ± 1.0 | 0.2 | −2.0 ± 1.0 | 1.3 | −1.5 ± 0.9 |
| Grassland | 1.0 | −1.3 ± 1.0 | 0.3 | −2.0 ± 0.8 | 2.1 | −1.5 ± 0.9 |
| Pasture/Hay | 2.1 | −1.1 ± 0.9 | 1.8 | −2.0 ± 0.5 | 0.4 | −2.2 ± 1.3 |
| Cultivated Crops | 16.8 | −1.7 ± 1.3 | 36.2 | −2.5 ± 1.4 | 5.2 | −1.3 ± 0.7 |
| Wetlands | 44.5 | −1.5 ± 1.2 | 35.5 | −2.6 ± 1.4 | 61.6 | −1.7 ± 1.0 |

GA-SC-NC refers to the coast of Georgia (GA), South Carolina (SC), and North Carolina (NC). The uncertainty is obtained as the standard deviation associated with the InSAR vertical land motion (VLM) obtained using Eq. (8).

affected by subsidence account for most of the exposed NLC (Fig. 4a–c). On the coast of Georgia (GA), South Carolina (SC), and North Carolina (NC) (referred to as GA-SC-NC throughout) the major subsiding cities are Brunswick (GA), Savannah (SC), and Charleston (SC), with rates exceeding 4 mm per year (Fig. 4d–f). The major NLC undergoing subsidence in the region are wetlands and forests, accounting for 88.3% of all NLC (61.2% for wetlands and 27.1% for forests) (Fig. 4f and Table 1).

### Coastal wetland vulnerability

Subsidence is a major threat to wetlands across the US Atlantic coast and is rarely fully accounted for in estimates of wetland vulnerability[41,43]. We calculated the vertical vulnerability of marsh pixels along the U.S. Atlantic coast using the vertical resilience (VR) index. The VR is a standard index used to estimate an accretion deficit (i.e. wetland accretion rate minus the relative SLR)[38,63,64]. Marshes with VR values greater than 0.5 mm per year were considered aggrading relative to SLR and thus not vulnerable to contemporary SLR. VR values less than −0.5 mm per year indicate marshes that are becoming more inundated and therefore vulnerable to SLR, while VR values between −0.5 mm per year and 0.5 mm per year indicate marshes that have largely accreted to keep pace with SLR[64] (see methods). To integrate our estimates of VLM and their uncertainties into estimations of wetland VR, we modified the VR formulation as the VLM minus SLR and

created a 90% confidence interval for VR (Eq. 9). The assessment of wetland vulnerability using InSAR provides a holistic measure of both surface elevation gain or loss due to sediment accumulation and compaction, incorporating both effects of deep and shallow subsidence on wetlands[43,45]. Additionally, unlike point estimates of accretion rate, which are measured at discrete locations[41] and sometimes extrapolated across entire watersheds[38], InSAR-based VLM provides a semi-continuous measurement of the accretion rate of each point along the wetland with a pixels dimension of ~50 m. We further correct the tide gauge measurements for the effect of local VLM by using observations from nearby GNSS stations. Because marsh accretion and vertical vulnerability can be highly dependent on marsh elevation[64], we also distinguished between low and high-elevation marshes utilizing elevations normalized to mean high water[65] (see methods section).

The VR maps for low- and high-elevation wetlands across the US Atlantic coast are shown in Fig. 5a–d. The analysis indicates that 57.6–100% of marshes are losing elevation relative to SLR (Fig. 5 and Supplementary Fig. 6), with little dependence on existing marsh elevation (low- or high-elevation). In contrast, we observe accretion deficits in only 43% (low-elevation) and 53% (high-elevation) marshes based on previously compiled accretion rates from a limited number of discrete points[38,64] (Supplementary Fig. 7a, b). By resampling VLM rates for the sites with accretion data, we show the discrepancy is due primarily to the inclusion of subsidence in the VLM data and is not sensitive to marsh elevation (Supplementary Fig. 7c, d). Therefore, our results suggest unprecedented marsh vulnerability on the US Atlantic coast that is directly linked to large-scale subsidence (Fig. 2c).

## Discussion

Long-term and immediate effects of SLR include increased flooding, saltwater intrusion into surface waters and groundwaters, and the decline of coastal wetlands and marshes[66]. However, in some places, the contribution from land subsidence could be ten times greater than that of the global mean SLR[67]. The combined effect of land subsidence and SLR increases socioeconomic and ecosystem exposure to coastal hazards[10,13,21,51,66]. Our spatially semi-continuous VLM rate map for the US Atlantic coast presents unprecedented spatial and high-precision data for the region. Given ongoing subsidence on the US Atlantic coast, the major NLC types with the most significant exposure include developed regions, cultivated crops, forests, and wetlands.

The Atlantic coast of the US has an estimated population of over 118 million, with some of the most populated cities in the US such as Boston (MA), New York City (NY), Norfolk (VA), Charlotte (NC), Baltimore (MD), and Miami (FL) (Fig. 2c). Several of these cities are already notable areas of persistent flooding[30,34,35,68] (Fig. 1), and ongoing subsidence rates exceeding 3 mm per year will amplify future inundation.

Agricultural lands along the US Atlantic coast show some of the highest rates of subsidence in all measured NLC types. Subsidence affects agricultural lands globally, likely due to the extensive groundwater overdraft for irrigation and subsequent compaction of aquifers[50,69,70], or the oxidation of organic materials[71,72]. For low-elevation agrarian areas around the coasts, the effect of subsidence on agricultural lands is both immediate and transformative, causing increased saltwater intrusion, storm surges flooding, and inundation, which leads to imbalances in nutrient concentrations, dead soils, and eventually the loss of agricultural lands[73]. The loss of agricultural lands would have a significant, far-reaching impact on the economy of the affected regions.

The observed subsidence of forest land may also contribute to the rapid expansion of 'ghost forests' along the US Atlantic coast through the impacts of saltwater intrusion[74,75]. Rates of coastal forest retreat are directly linked to the rate of relative SLR[75] and are leading to fundamental ecological shifts[75–79]. More than 8% of forested coastal wetlands have been displaced along the North American Atlantic coastal plain[80],

and forest retreat has led to associated increases in marsh area[81]. Among the implications of that change is the loss of carbon stored in the aboveground biomass of trees[75,82], which is only partially compensated by the carbon stored in developing marsh soils[82].

The response of marshes to SLR is hotly debated, and based in part on their ability to accrete vertically at a rate equivalent to relative SLR[4,64,82–85]. Here, we show through measurements of VLM across the entire Atlantic Coast that the current balance between VLM and relative SLR is negative in 58–100% of marshes (754–1300 km²). These measurements are consistent with observations of nonlinear increases in subsidence with accretion that amplify march vulnerability (e.g., ref. 45), and localized observations of ecological shifts toward flood-tolerant vegetation where accretion deficits exist (e.g., ref. 86). However, we do not account for future accelerations in either the rate of SLR or vertical accretion. Nevertheless, our estimates of VLM uniquely show that subsidence can tip the accretionary balance of marshes towards submergence, and that vulnerability assessment that does not fully account for subsidence will underestimate marsh vulnerability.

Our current vulnerability estimates are valid under current rates of subsidence and SLR and the upper bounds of the vulnerability estimates may represent the worst-case scenario for wetlands on the US Atlantic coast. However, considering future projections of relative SLR (e.g., accelerated rates of SLR under various representative concentration pathway (RCP) scenarios) may exacerbate the wetland vulnerability estimates with a significant impact on the entire coastal ecosystem.

Accurately accounting for the contribution of VLM and the drivers of wetland loss are essential for understanding and predicting coastal inundation hazards. Local rates of relative SLR often vary vastly from regional SLR. This disparity is influenced mainly by regional ocean dynamics and the strong contribution of local land subsidence to eustatic SLR. However, current projections of relative SLR underestimate/overestimate the contribution of VLM (Fig. 3), which undermines the reliability of inundation models. Our findings emphasize the role of InSAR VLM measurements in refining local and regional rates of relative SLR and ensuring the accuracy of coastal inundation models. Wetlands—nature's flood control structures and erosion buffers—protect large sections of the coast[5,27,87]. Notably, many of the world's coastal megacities are located on river deltas, which are subsidence centers; sinking faster than the rising seas (e.g., Mekong (Vietnam), the Mississippi (USA), Niger (Nigeria), Nile (Egypt), Ganges Brahmaputra (India/Bangladesh), ChaoPhraya (Thailand), and Yangtze (China))[88]. In these environments, the resilience of human habitats is inextricably linked to the survival and preservation of wetland ecosystems[89]. Thus, high-resolution estimates of VLM must be incorporated into wetland survival metrics to ensure the effective implementation of sustainable coastal adaptation strategies.

## Methods

### SAR analysis

The SAR datasets contain 3057 images acquired in ascending orbit geometry spanning 2015–2020 for the Sentinel-1 datasets and 2007–2011 ALOS datasets. The datasets contain 15 and 81 frames for the Sentinel-1 and ALOS satellites, respectively (Fig. 1 and Supplementary table 1). To minimize the errors associated with InSAR wetland monitoring and to maximize the pixels over wetlands, we screened the SAR (Sentinel-1) images by implementing a statistical framework for flood mapping[90], prioritizing images obtained during low tides; with the maximum exposed surface. For the Sentinel-1 dataset, we generated 7005 interferograms using maximum temporal and perpendicular baselines of 500 days and 700 m, respectively. While for the ALOS dataset, we generated 13,055 interferograms with maximum temporal and perpendicular baselines of 1500 days and 2500 m, respectively. The Sentinel-1 and ALOS data combination allows for a broader temporal spread of data and produces a more robust result.

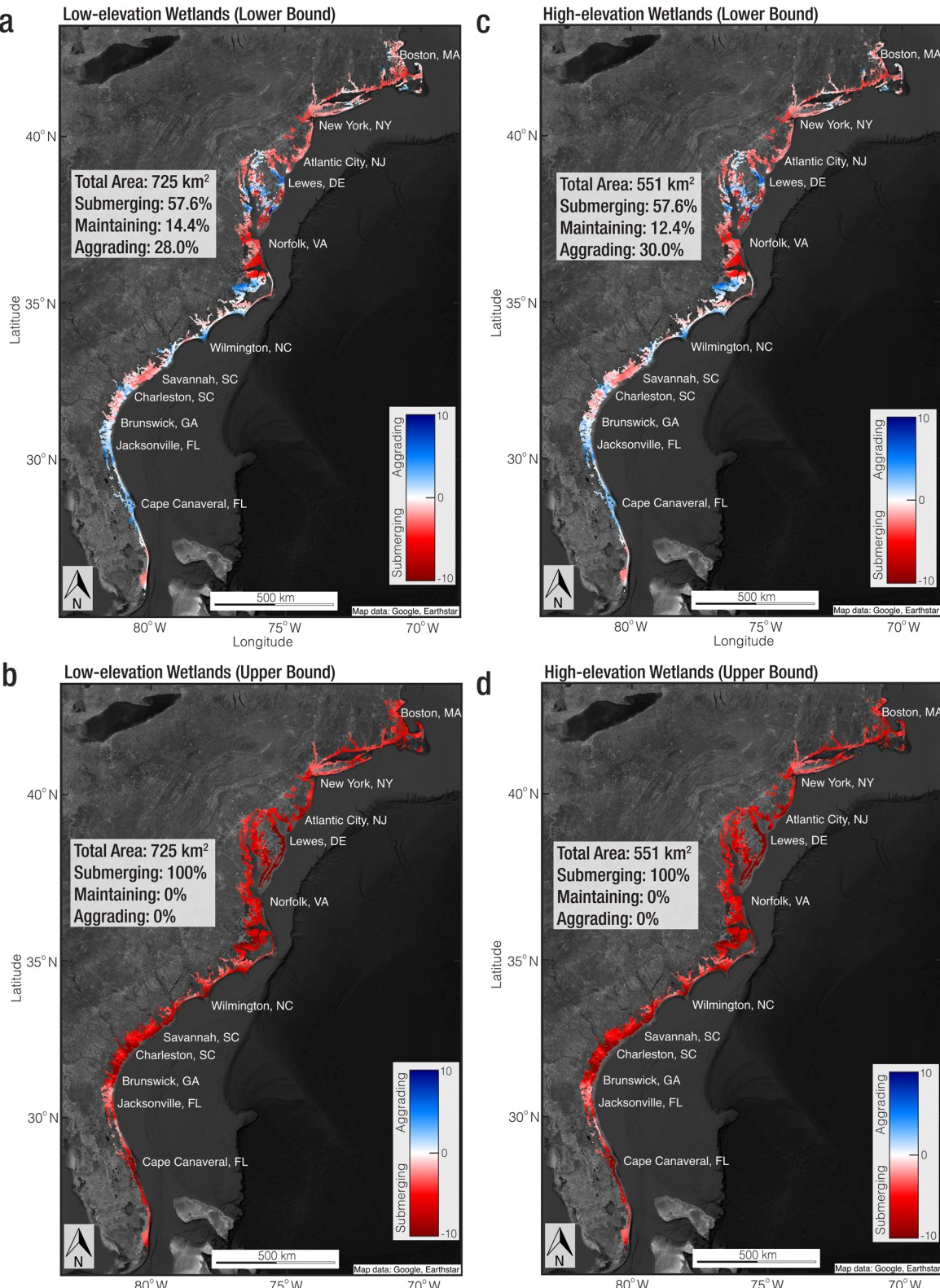

**Fig. 5 | Wetland vulnerability across the US Atlantic coast. a** Lower bound of vertical resilience (VR) for low-elevation wetlands (Background Image: Google, Earthstar). **b** Upper bound of VR for low-elevation wetlands (Background Image: Google, Earthstar). **c** Lower bound of VR for high-elevation wetlands (Background Image: Google, Earthstar). **d** Upper bound of VR for high-elevation wetlands (Background Image: Google, Earthstar). VR is calculated as vertical land motion (VLM) minus sea level rise. The upper bounds and lower bounds are calculated as ±

2 SD. Red colors indicate submerging wetlands (VR < −0.5 mm per year), blues indicate aggrading wetlands (VR > 0.5 mm per year), and whites indicate maintaining wetlands (−0.5 mm per year ≤ VR ≤ 0.5 mm per year). Note that the scatter plots are exclusive to wetland areas, whereas the large-scale map provides the illusion of non-wetland pixels by magnifying each point. A close-up map of the wetland vulnerability in the Chesapeake Bay region is provided in Supplemental Fig. 6.

We applied the multitemporal wavelet-based InSAR (WabInSAR) algorithm to generate high-resolution time series of deformation for each SAR frame[52,53,91,92]. The wavelet-based analyses applied to the interferograms involved identifying and removing noisy pixels, reducing the effects of topographically-correlated atmospheric phase delay, spatially uncorrelated DEM error[52,53] and ionospheric errors[93]. The geometrical phase was calculated and removed using the 30-m Shuttle Radar Topography Mission (SRTM) digital elevation model[94] and the satellite's precise ephemeris data[52]. Next, the average interferometric coherence and time series of the complex interferometric phase noise were analyzed to identify the elite pixels[53]. The so-called elite pixels are those with an average coherence larger than 0.7 and a normally distributed temporal interferometric noise[53]. We obtained the absolute estimate of the phase change for elite pixels via an iterative 2D sparse phase unwrapping algorithm. Each unwrapped interferogram was corrected for the effect of orbital error[95] and the topography-correlated component of atmospheric delay[52]. Using a robust regression that implements a reweighted least square, we inverted the phase changes from the interferograms[95]. We applied a high-pass filter, using continuous wavelet transforms to reduce the temporal component of the atmospheric delay. We then calculated each elite pixel's velocities along the LOS direction as the slope of the best-fitting line to the associated time series using a reweighted least-squares estimation. Lastly, we geocoded all data sets to obtain precise locations of elite pixels in a geographic reference frame. The final LOS velocity for each frame is mosaiced to generate two large-scale maps of LOS displacements for Sentinel-1 and ALOS datasets[96]. We transform the LOS velocities from local to a global reference frame by using an affine transformation[51] to mitigate error propagation associated with the mosaic process and reduce the long-wavelength errors and residual orbital errors. The final locations of elite pixels for Sentinel-1 and ALOS datasets and 173 GNSS stations used in this study are shown in Supplementary Fig. 8a, b. For each GNSS station, we selected only stations with at least 50 measurements yearly between 2007 to 2020 for our analysis (Supplementary Fig. 9). Supplementary Fig. 1a, b shows the final LOS velocity for both Sentinel-1 and ALOS datasets.

The obtained LOS velocity comprises projections of 3D displacement velocity on Sentinel-1 and ALOS satellites' LOS directions. Consequently, we devise an approach to decompose the LOS velocity in combination with GNSS observations, into horizontal and vertical components, with the east (E), north (N), and vertical (U) velocities. To this end, we first resampled the LOS velocities of the Sentinel-1 dataset on the location of pixels within the ALOS dataset using the nearest neighbor algorithm. Next, we implemented a Kriging interpolation approach with inverse distance weighting to interpolate 132 GNSS horizontal and vertical velocities (76% of all GNSS observations) on the location of elite pixels within the ALOS dataset. This procedure provides five observations per pixel, including two LOS observations and three GNSS velocities.

Assume $\{y_{alos}, y_{sen}\}$ and $\{\sigma_{alos}^2, \sigma_{sen}^2\}$ as the interpolated LOS velocities and the variances for a given pixel, where the subscripts *alos* and *sen* indicate ALOS and Sentinel-1, respectively. The stochastic model to combine the LOS velocities with the GNSS datasets to generate a high-resolution map of the E, N, and U velocities are given by Eq. (1):

$$
\begin{aligned}
y_{alos} &= C_e^{alos}E + C_n^{alos}N + C_u^{alos}U + \varepsilon^{alos}\\
y_{sen} &= C_e^{sen}E + C_n^{sen}N + C_u^{sen}U + \varepsilon^{sen}\\
E_{GNSS} &= E + \varepsilon^e\\
N_{GNSS} &= N + \varepsilon^n\\
U_{GNSS} &= U + \varepsilon^u
\end{aligned}
\tag{1}
$$

Where $C$ represents the unit vectors projecting 3D displacements onto the LOS and is a function of the heading and incidence angles, $\varepsilon$ is the

observation error equal to the standard deviation ($\sigma$), which is assumed to be normally distributed. From Eq. (1), E, N, and U are unknown and differs from $E_{GNSS}$, $N_{GNSS}$, $U_{GNSS}$ which are the observed interpolated east, north, and up GNSS velocities. We assigned a $\sigma$ of 3 mm per year and 1 mm per year for the ALOS and Sentinel LOS velocities, respectively based on previous studies[51,96]. We used the provided error by Nevada Geodetic Laboratory[97] for the GNSS velocities. This stochastic model is known as a unified weighted least-squares adjustment and can be represented in matrix form in Eq. (2), and the solution is given by Eq. (3):

$$
\begin{pmatrix} y_{alos}\\ y_{sen}\\ E_{GNSS}\\ N_{GNSS}\\ U_{GNSS} \end{pmatrix} = \begin{pmatrix} C_e^{alos} & C_n^{alos} & C_u^{alos}\\ C_e^{sen} & C_n^{sen} & C_u^{sen}\\ 1 & 0 & 0\\ 0 & 1 & 0\\ 0 & 0 & 1 \end{pmatrix} \begin{pmatrix} E\\ N\\ U \end{pmatrix}
\tag{2}
$$

$$
X = \left(G^T P G\right)^{-1} G^T P L
\tag{3}
$$

Where $X$ represents the unknowns, $G$ is the Green's function, $L$ is the observations, and $P$ is the weight matrix, which is inversely proportional to the observant variance ($\sigma^2$).

$$
\begin{aligned}
L &= (y_{alos}\, y_{sen}\, E_{GNSS}\, N_{GNSS}\, U_{GNSS})^T\\
X &= (E N U)^T\\
G &= \begin{pmatrix} C_e^{alos} & C_n^{alos} & C_u^{alos}\\ C_e^{sen} & C_n^{sen} & C_u^{sen}\\ 1 & 0 & 0\\ 0 & 1 & 0\\ 0 & 0 & 1 \end{pmatrix}\\
P &= \begin{pmatrix} \sigma_{alos}^{-2} & 0 & 0 & 0 & 0\\ 0 & \sigma_{sen}^{-2} & 0 & 0 & 0\\ 0 & 0 & \sigma_e^{-2} & 0 & 0\\ 0 & 0 & 0 & \sigma_n^{-2} & 0\\ 0 & 0 & 0 & 0 & \sigma_u^{-2} \end{pmatrix}
\end{aligned}
\tag{4}
$$

To assign a weight to interpolated GNSS values, we use the following relation that assigns to each pixel a weight proportional to that of its nearest GNSS station, where $D$ is the distance in km and $s_e$, $s_n$, and $s_u$ are the interpolated SD of the east, north, and up GNSS velocities, respectively. The denominator value of 10 is the average distance between GNSS stations. We chose the uncertainty to increase with distance, tested other values, and found that a denominator of 10 yields a 3D displacement field that best fits independent GNSS measurements.

$$
\begin{aligned}
\sigma_e &= s_e * (1 + \tfrac{D}{10})\\
\sigma_n &= s_n * (1 + \tfrac{D}{10})\\
\sigma_u &= s_u * (1 + \tfrac{D}{10})
\end{aligned}
\tag{5}
$$

The E, N, and U velocities are shown in Supplementary Fig. 2 and Fig. 2.

## Error analysis and validations

Here we examine the quality of our results in terms of precision (i.e., stand deviation) and accuracy (i.e., closeness to the true value). To this end, we first employ the concept of error propagation[98] to obtain parameter's variance-covariance matrix, given observation errors. The combination of ALOS, Sentinel-1, and GNSS data increases the redundancy and enables adjusting the observation errors[51]. Thus, the

parameter variance-covariance matrix[98] is given by Eq. (6):

$$Q_{XX} = \frac{r^T \mathrm{Pr}}{df}\left(G^T P G\right)^{-1} \qquad (6)$$

where df is the degrees of freedom = 2 and r are the residuals given by Eq. (7):

$$r = L - GX \qquad (7)$$

Supplementary Fig. 3a–c shows the SD of each pixel for the E, N, and U velocities evaluated from Eq. (6). The spatial distribution shows that most SD values are less than 3 mm per year (Supplementary Fig. 3). However, the SD around the Chesapeake Bay is relatively higher. A higher SD value reflects the reliability of reported VLM rates and can be due to the limited number of GNSS stations used in the adjustment, the relatively higher SD of the available GNSS station in the region, and variations in the rate of surface deformation between ALOS and Sentinel-1 periods[51].

Next, we perform additional validation tests to assess the accuracy of our results. To this end, we compared the derived horizontal and vertical velocities with observations from 41 independent GNSS stations selected randomly, as ground truth. The comparison was done by averaging the measurements of pixels within a radius of 200 m for each GNSS station used in the validation. Next, we evaluate the difference between InSAR and GNSS estimates of 3D displacement field. We found a SD of 0.22, 0.18, and 1.28 mm per year for the differences in E, N, and U velocities, respectively (Supplementary Fig. 2b, d and Fig. 2b). The mean of the differences is insignificant and near zero.

The reported uncertainties or SD throughout the study (the InSAR VLM projections at tide gauges and NLC) are averaged standard deviations associated with InSAR pixels obtained using Eq. (8):

$$s = 1/n \sqrt{\sum_i^n SD_i^2} \qquad (8)$$

Where $SD_i$ is the ith InSAR pixel standard deviation derived using Eq. (6) and n is the number of pixels within 200 m of each GNSS or tide gauge being averaged, or those associated with the VLM of the NLC type.

The east, north, and VLM rates and standard deviations are avialble at https://doi.org/10.7294/19350959.

## National land cover data
The land cover data were obtained from the 2019 National Land Cover (NLC) Database managed by USGS[29]. The NLC database provides 30-meter resolution data on land surface characteristics and land surface change at the Landsat Thematic Mapper (TM). The NLC data for the US east coast has 15 principal features, viz open water, developed (open space, low intensity, medium intensity, high intensity), barren land, forest (deciduous forest, evergreen forest, mixed forest), shrub/scrub, grassland, pasture/hay, cultivated crops, and wetlands (woody wetlands, emergent herbaceous wetlands). We combined VLM velocity with the NLC by resampling the VLM velocity for the different NLC and extracted the VLM corresponding to each NLC pixels. Next, we simplified the NLC by classifying NLC with similar characteristics to obtain only 8 NLC categories (developed, barren land, forest, shrub/scrub, grassland, pasture/hay, cultivated crops, and wetlands).

## Wetland vulnerability
We analyzed wetland vulnerability using vertical resilience to SLR. To characterize the vertical resilience of wetlands, we first obtained the relevant data, including the point accretion rate, SLR, and the location of wetlands for the US Atlantic coast. The accretion data included 182

data points from Georgia to New Hampshire obtained by combining the reported accretion rates for the US Atlantic coast from Holmquist et al.[38] and Kirwan et al.[64] (Supplementary Fig. 7e). We screened the accretion data to only include 137 rates based on $^{137}$Cs-dated cores and horizon markers. The Tide gauge measurements were obtained from the National Oceanic and Atmospheric Association (NOAA) sea level trends, datum periods span 1900 to 2020 (ref. 99) and corrected for VLM to obtain absolute SLR using the median up the velocity of GNSS stations within a radius of 800 m (Supplementary Fig. 10a, b). We compiled the wetland data by mapping all wetlands in the US Atlantic coast from the 30 m grid map of relative tidal marsh elevation provided by Holmquist & Windham-Myers[65]. This map is a comprehensive database of wetland elevation across the conterminous US normalized to mean high water (MHW), which combines estuarine and freshwater wetlands from NOAA Coastal Change Analysis Program (C-CAP) and tidal wetlands from the National Wetlands Inventory (NWI).

We adopted the Holmquist & Windham-Myers[65] elevation normalized to MHW ($Z_{MHW}$) to differentiate between low- and high-elevation wetlands. We implemented $Z_{MHW} = 1.0$ as a reasonable criterion to differentiate low- and high-elevation wetlands, according to the recommendation by Holmquist & Windham-Myers[65]. High-elevation wetlands are characterized as infrequently inundated, while and low-elevation wetlands are frequently inundated. We thus defined wetlands with $Z_{MHW} \geq 1.0$ as high-elevation wetlands and $Z_{MHW} < 1.0$ as low-elevation wetlands. The $Z_{MHW}$ for the 137 accretion data shows a broader range of accretion rates for $Z_{MHW} < 1.0$ than $Z_{MHW} \geq 1.0$ (Supplementary Fig. 11a). However, the VLM rates at locations with accretion data measurements displayed no identifiable characteristic to distinguish low- and high-elevation wetlands (Supplementary Fig. 11b). To obtain the VLM and SLR for the wetlands in the US Atlantic coast, we resampled the VLM rate on the wetland pixels, discarding pixels ≥10 m away from the InSAR pixels and interpolated the corrected SLR data on the wetland pixels. The final wetland map contained ~1.4 million pixels, covering approximately 1300 km². Using elevation rate data from Saintilan et al.[45], we attempted validation of the VLM data over wetlands. The elevation rate data for wetlands was collected using SET-MH stations that incorporates both accretion rate and subsidence on wetlands. For the comparison, we only included stations within a 10 m radius of InSAR pixels consistent with the radius for wetland pixel selection. Only two of the fourty-three SET-MH stations on the US Atlantic coast were located within a 10-meter radius of the InSAR pixels. This suggests that the site prioritization of SET-MH monitoring stations and the location of InSAR wetland pixels are different. These disparities may be due to the fact that SET-MH monitoring stations are potentially placed in permanently/frequently flooded zones, as their locations are influenced by human apriori knowledge. In contrast, InSAR pixels are based on wetland surface exposure and favor less inundated areas. However, InSAR data represents the greater number of subsamples of wetlands on the US Atlantic coast compared to point SET-MH measurements. The comparison of the SET-MH elevation change rates to the InSAR VLM rates is shown in supplementary table 3 and shows consistency between the measurements.

We calculated the vertical resilience (VR) as the VLM minus SLR (Eq. 9), modifying the standard equation of net accretion (RSLR subtracted from accretion). To account for observation uncertainties in the VR estimates, we used 2 times the SD of VLM rate to establish a 90% confidence interval (Eq. 9) and obtain a lower and upper bound on VR. A VR value larger than 0.5 mm per year indicates aggrading wetlands, while a VR value less than −0.5 mm per year indicates submerging wetlands. A VR value between −0.5 mm per year and 0.5 mm per year indicates wetlands that can keep pace with SLR. This threshold was necessary to account for uncertainties in the measurements, the dependence of sea level rates on the recording period, and the potential for short-term marsh accretion rates to fluctuate with

historical SLR rates[64].

$$VR = VLM - SLR$$
$$VR_{lowerbound} = VLM + 2SD - SLR \qquad (9)$$
$$VR_{upperbound} = VLM - 2SD - SLR$$

The upper and lower bounds of VR calculated at 90% confidence range for low- and high-elevation wetlands following Eq. (9) are shown in Fig. 5a–d. Supplementary Fig. 7a, b shows the VR calculated as accretion rate minus the relative SLR for the pointwise aggradation data, while Supplementary Fig. 7c, d shows the VR calculated for pointwise data following Eq. (9).

## Data availability

The east, north, VLM rates, and standard deviations data generated in this study have been deposited in the figshare database [https://doi.org/10.7294/19350959]. The global navigation satellite system (GNSS) velocity data used in this study are available from the Nevada Geodetic Lab [http://geodesy.unr.edu]. The Synthetic Aperture Radar (SAR) datasets used in this study are available from the Alaska Satellite Facilities [www.asf.alaska.edu]. The national land cover (NLC) data used in this study are available from the U.S. Geological Survey [https://www.usgs.gov/centers/eros/science/national-land-cover-database]. The global multi-resolution topography (GMRT) used in in Fig. 4 was made with GeoMapApp [www.geomapapp.org]. The tide gage data used in this study are available from the National Oceanic and Atmospheric Administration [https://tidesandcurrents.noaa.gov]. The wetland normalized to mean high water data used in this study are available from the Oak Ridge National Labs Distributed Active Archive Center [https://doi.org/10.3334/ORNLDAAC/1844]. The groundwater data used in this study presented in supplementary Fig. 5 is available from the U.S. Geological Survey water data [https://waterdata.usgs.gov/nwis]. Source data are provided with this paper.

## Code availability

The WabInSAR code used to perform the synthetic aperture radar (SAR) analysis is available at https://sites.google.com/vt.edu/eadar-lab/software?authuser=0.

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

## Acknowledgements

L.O. and M.S. are supported by NASA Grant (80NSSC170567) and a grant from US Geological Survey. M.K. is supported by the National Science Foundation (#1654374, #1832221, and #2012670).

## Author contributions

M.S. designed the study. L.O., M.S., and C.O. performed the synthetic aperture radar (SAR) analysis. L.O. and M.S. created the figures and wrote the first draft of the paper. L.O., M.S., and M.K. performed the vulnerability analysis. L.O., M.S., C.O., and M.K. analyzed the data and edited the manuscript.

## Competing interests

The authors declare no competing interests.
