## [Peer Review File · Nature Communications]

Hidden Vulnerability of US Atlantic Coast to Sea-level Rise due to Vertical Land MotionEditorial Note: Parts of this Peer Review File have been redacted as indicated to remove third-party material where no permission to publish could be obtained.

REVIEWER COMMENTS

Reviewer #1 (Remarks to the Author):

Ohenhen et al., provide an interesting and important analysis of VLM rates along the US Atlantic Coast. Their findings regarding wetland vulnerability are particularly notable given the lack of certainty in the community regarding the ecosystem's ability to adapt to sea-level rise. The article is overall well written - minor comments are highlighted in the attached pdf.

A bit more severe concern is the general ability of the figures to present relevant information. In figure 1, I do not see the purpose of the mean flood days increase. How these are derived is not discussed. What geophysical processes are included in the 'coastal waters', what a 'local threshold', and even where this is occurring is unclear. My guess is that you are taking a point located at a tide gauge (e.g., Charleston) and then propagating this to say the whole city is experiencing more days in recent years. As this paper highlights the importance of spatial variability, this would not seem to be a consistent message.

Figure 2a is essentially just a wash of yellow. The subsidence panel on the right does little to help. Is all of NC, except for the few red hotspots, subsiding at the same rate? That yellow rate looks anywhere between -1 and -3 mm/yr, which is a significant difference given the range of magnitudes. You need to figure out a different method (e.g., a discrete colorbar) to bring out the features. More importantly, it does not seem to agree with Figure 4b, which should be the same data. Specifically, there is no data in the southwest corner of the Delmarva peninsula in 4b but there does seem to be data there in Fig 2.

Figure 3 is flat out missing uncertainties in several places. The units are a) inconsistent with the rest of the paper and b) increase the difficulty of interpretation. Also, the projection analysis seems tangential to the paper and is poorly discussed. What justification is there to extend 13 InSAR rates 78 years into the future? How do the methodologies of the IPCC VLM and InSAR compare?

Figure 4: It is unclear how VLM pixels are tied to the land cover classes. Please elaborate on this sentence in the methods:

1. "We combined VLM velocity with the NLC by resampling the VLM velocity for the different NLC."

Figure 5: Why do you correct the TG with GPS instead of the InSAR? This seems to defeat the purpose (and is inconsistent with the method used in Figure 3). Additionally, the nearest GPS to TG can be quite far and unrepresentative of actual VLM.

Most seriously, I do not see sufficient evidence to trust the generated VLM rates. The underlying LOS dataset is not reproducible. There are myriad community tools for InSAR processing and analysis that are well established, widely vetted, and carefully documented (e.g., ISCE, MintPy). The WabInSAR (Shirzaei, 2013) used has none of those properties and is not available to the community (as far as I am aware). As such, this contradicts Nature's commitment to open science (<https://www.nature.com/nature-portfolio/editorial-policies/reporting-standards>) and falsifies the authors' statement in the `Data and materials availability` section.

While the theoretical basis of WabInSAR is documented and shown to perform well for a particular sensor and study area with substantial displacement (Shirzaei, 2013), InSAR algorithm performance varies widely given the number of error sources that vary spatiotemporally. This is especially relevant given the mm-scale displacement (vs centimeter scale in Shirzaei, 2013). However, no discussion of error sources and minimal uncertainty metrics are presented in the submission.

Uncertainties are primarily reported regarding the fit to GPS stations. While these numbers look good enough, there are only a relatively small number left out for validation (41/460) permitting overfitting.

More importantly, the GPS are primarily constraining the long wavelength signals. In between GPS stations, short wavelength InSAR displacement signals are subject to noise. Indeed, how many GPS stations are in wetlands?

This is especially important given the challenges of InSAR in wetlands, which decorrelate when inundated, and are almost by definition in vegetated areas (which are also susceptible to decorrelation). In addition to adding timeseries noise, decorrelation leads to challenges in unwrapping. How are these addressed? What is the temporal coherence of the wetlands? What do some timeseries look like?

Where uncertainties are reported, it's ambiguous as to what they even are. The data file and Figs 3, 5, Table 1, Eq 6, Fig S3, show a standard deviation associated with the VLM rate. Is this the standard deviation of the time series? The standard deviation of bootstrapped rates? A measure of the residue of the linear fit? How does it compare to the rates? The colors makes it impossible to tell.

How do localized results compare to existing results (e.g., Bekaert et al., 2017, Scientific Reports)?

Can you provide the data in a more usable and realistic way? Why are these point measurements if they are 50m spatial resolution? A raster would be more appropriate and much easier to work with.

Given these considerations, I do not think this article warrants publication in Nature Communications.

Reviewer #2 (Remarks to the Author):

There is an ongoing debate in the literature regarding the relative contribution of subsidence to RSLR, particularly on the US Atlantic coast. In this paper, the authors use SAR, GNSS, and tide gauge data to produce coast-wide measurements of vertical land motion and subsidence. Results provide strong evidence that wetlands on the Atlantic coast are indeed experiencing widespread subsidence, and as a result, coastal wetlands are increasingly vulnerable to inundation. With minor revisions, this will be an exciting contribution to the literature.

Line 81: Clarify that although many assessments of coastal vulnerability do not account for both shallow and deep subsidence, the studies you cite here are ones that DO. May want to mention a few studies that do not account for shallow subsidence.

Line 99: What is the date range for the GNSS observations?

Line 128: The subsidence rate in NYC appears to be ~1.5 mm/yr. This is certainly not something to ignore, but it doesn't jump out at me as a hotspot of subsidence. Boston appears to be subsiding faster (~3 mm/yr). Or maybe you are simply listing large Atlantic coast cities, most of which are undergoing subsidence?

Lines 129-130: I do not see that subsidence is high in the Chesapeake Bay region as a whole. It appears that areas around Delaware Bay and the eastern and southern Delmarva Peninsula are subsiding rapidly, but much of the area around the Chesapeake Bay has little subsidence or is uplifting (as you note later in this paragraph).

Lines 137-138: Are the IPCC's VLM rates based on tide gauge data?

Lines 140-144: What is your buffer for an "accurate" measurement of VLM (not an over or underestimate)?

Line 149: How are you defining "exposure" to subsidence?

Lines 153-156: This sentence implies that some land cover types experience subsidence while others do not, but these four categories account for 96% of all land cover types in the study area. I suggest simply stating the four categories and corresponding percent land cover (i.e., cut "Subsidence exposure follows dominant land cover, where").

Lines 160-161: Yikes, 11.8 mm/yr of subsidence. Where is this?

Line 172: By "agrarian land", do you mean cultivated crops plus pasture/hay?

Line 192: As you mention below, VLM as measured by InSAR captures all subsidence processes (shallow and deep). Tide gauges, however, typically do not record shallow subsidence. Thus, differencing these two types of data will give you an overestimate of VR (because shallow subsidence is not subtracted out). The inclusion of shallow subsidence in SLR rates would likely indicate that more regions are experiencing a VR deficit, or a greater deficit. Please make note of this in the text.

Line 200 (and methods): Typically, GNSS do not measure shallow subsidence either, so they do not solve the overestimation of VR mentioned above.

Lines 211-213: Very clear summary of results. Note that vulnerability values would be even greater if shallow subsidence measurements were included in SLR!

Lines 216-224: Nice summary of the importance of this work.

Lines 238-240: These two sentences are duplicated in the results. Cut from one of these sections.

Figure 1: Excellent figure. Packs in a lot of information without feeling cluttered or confusing. Very nice. Note that the study area is the US Atlantic coast from New Hampshire to Florida (excluding Maine), so not quite the whole US coast.

Figure 2: How are panels a and c different? Is c just the negative (subsidence) component of a? That is fine, but clarify in the caption.

Figure 3: Please make all text larger! Is the blue shaded region (2 SD) present in each panel but too narrow to see in some cases (Woods Hole, Bridgeport, etc.)? If so, note this in the caption.

Supplemental Figure 5: The blue triangle corresponds to the red curve and the red triangle corresponds to the blue curve? Please fix.

Reviewer #3 (Remarks to the Author):

ARTICLE SUMMARY

Vertical land motion (subsidence vs uplift) remains poorly monitored globally, despite being necessary to calculate relative sea level change at the coast accurately. This is particularly worrying, given the dual threat of accelerating eustatic sea level rise and subsidence (often linked to anthropogenic activity) that is increasingly inundating coastal cities, forests, agricultural land, and wetlands. To address this important knowledge gap, the authors used satellite-derived data to map vertical land motion rates across the US Atlantic coast, and show that most of the coastline is subsiding. For saltmarshes in particular, the authors estimate that between 58 - 100% of marshes are losing elevation upon calculating the difference between local sea level rise and vertical land motion rates.

The authors point to an even greater risk social, economic, and ecological systems faced by the US Atlantic coast now that subsidence has been mapped for the first time. Moreover, vertical land motion rates used by the IPCC differ from the author's calculations, indicating greater uncertainty in current assessments of future coastal flood risk. The authors advocate for the need to better account for subsidence when planning coastal adaptation strategies.

GENERAL COMMENTS

Including vertical land motion in assessing coastal flood risk remains an important research gap, especially in the context of saltmarshes where a small difference in relative sea level rise rate can define whether they survive or drown. If up to 100% of marshes across the US Atlantic coast are indeed at risk of drowning due to subsidence, this represents an existential crisis for saltmarshes – especially if the US Atlantic coast situation is similar to other parts of the globe. This work overturns the current paradigm that marshes are expected to survive sea level rise due to a positive feedback between increased accretion and prolonged inundation by sea level rise. Moreover, the manuscript compliments the recent publication of Saintilan et al. (2022), who demonstrate from field observations that less than half of the sediment accreted on marshes is translated to surface elevation gain (made even worse by sea level rise, as the addition of more material increases shallow compaction). I therefore believe this manuscript is novel, timely, and of relevance to the wide readership of Nature Communications.

I would like to see the authors:

- State more explicitly in the text what the uncertainty in the vertical land motion rates (derived from InSAR data) are, how these were assessed, and what these mean for confidence in the reported subsidence rates. These points are covered throughout the text, but I believe they should be made easier for the reader to understand. For example, standard deviation is used as the uncertainty metric, and its derivation is described in lines 110-112 (for east and north velocities) and lines 133-134, 348-357 (from comparing InSAR upwards velocities to the GNSS data, also reported in Fig. 2). I'm unclear if both feature in the overall accuracy assessment. Moreover, do these measures account for the accuracy of the SAR sensors?
- Relate the relevance of their findings to other coastal environments beyond the US Atlantic coast – especially in the context of river deltas where subsidence is a key threat and often the result of development of the river catchment resulting in reduced freshwater flow and sediment flux to the coast.

The quality of writing is high, manuscript is formatted correctly, and the figures and supplementary materials are helpful and well presented. I believe the manuscript is robust and well referenced. Some points could be clearer, and I highlight these below.

SPECIFIC COMMENTS

Please refer to line numbers in the PDF version of this manuscript:

- Line 1: May I suggest two changes to the title: (1) "Increased" implies vertical land motion has recently begun to affect US Atlantic marshes, which is not the case. It's that vertical land motion has not been properly accounted for until now. "Hidden" or "Overlooked" may be more appropriate; (2) Since the effect of vertical land motion on forests, agricultural land, and cities are also being considered, I recommend removing "wetlands" from the title.
- Line 17: Given that cities are being considered too, is it better to replace "ecosystems" with "environments"?
- Line 18: remove "heterogeneous tidal ranges and land cover types" as the focus is on subsidence
- Line 20: since GNSS are also used in the analysis, maybe "satellite data" could be used instead of "synthetic aperture radar" as a catch-all term
- Line 24: The focus on coastal marshes in the sentence starting "We estimate..." leaves the reader wondering why wetlands forests / agricultural areas / developed regions aren't summarised. I would

add a connecting sentence to emphasise why wetlands become a key point

- Line 30: Is subsidence a climate-related risk? I would argue it is largely anthropogenically-induced
- Lines 41-42: "flooding" and "storm surge" are identified as separate hazards, though the former is often a consequence of the latter. Perhaps specify "river flooding" instead.
- Lines 45-47: For the example given here, I don't think the link between coastal ecosystems and coastal cities is particularly strong. I suspect many of the megacities function without much reliance on local coastal ecosystem services
- Line 47: "Coastal wetlands, which shelter most coastlines" is a bit of a sweeping statement. Coastal wetlands tend to occur in sheltered coastal settings
- Line 47: The statement that "coastal wetlands ... store more carbon ..." is no longer correct according to Temmink et al. (2022). Tropical peatlands hold store more carbon. Sequestration rates are generally higher in marshes, however. I would update reference 8 (McLeod, 2011) with Temmink et al. (2022).
- Line 51: "carbon sequestration" has been repeated from the previous sentence, so should be removed.
- Line 54: "(ref .14)" is used rather than the superscript form. These cases should be checked throughout the manuscript.
- Line 60: "...which threatens coastal ecosystems". Please summarise how subsidence could threaten coastal ecosystems
- Line 81: in addition to refs 37-40, Saintilan et al. (2022) should be referenced also
- Line 82: "InSAR overcomes these challenges" – please summarise how. E.g., "...by providing high spatial and temporal resolution on elevation change"
- Lines 110-111: "41 GNSS stations, which were not used in the analysis" – please explain why
- Line 130: I am unclear what is meant by "bounds" when discussing uplift in the Chesapeake Bay area
- Line 136: By "these measurements", do the authors mean the rate of sea level change? Please clarify
- Line 137: It would be helpful to know how the IPCC have projected rates of vertical land motion. Do they also use SAR or a different method?
- Lines 149-150: The implication here is that subsidence rates are embedded in the NLCs, which is incorrect. Please clarify
- Lines 153-154: "Subsidence exposure follows dominant land cover" – I'm unsure what is meant here. Please clarify
- Line 173: "This loss of agricultural land [due to subsidence]" – please clarify this somewhat. The land would be lost to salinisation, which increases due to increased coastal flooding because of subsidence, correct?
- Line 184: "a modification of" – since the modification isn't immediately explained, the reader is left wondering. I would remove this and allow the text later to explain the modification after VR is explained in the next sentence
- Line 187: The cut-off of 0.5 mm/yr chosen is not adequately explained. Please state why this cut-off was used.
- Line 189: is it that marshes are "vulnerable to SLR", or are they vulnerable to the effects of sea level rise (i.e. excessive inundation that exceeds the physiological tolerance of marsh plants)?
- Line 206: "with little dependence on existing marsh elevation" is unclear – please clarify
- Line 234: "the oxidation of organic materials" – please provide a reference
- Line 254: "...but do not account for future accelerations in..." - do the authors mean their work does not account for changes in the rate of SLR? Please clarify.
- Lines 259-261: This sentence could be clearer. I believe the authors are saying that accounting for VLM is needed to predict the true risk of coastal flooding.
- Line 273: "vulnerability estimates" – perhaps this is too generic. Do the authors specifically mean vertical land motion rates?
- Line 296: The SRTM has a vertical accuracy of +/- 16 m. Does this reduce its effectiveness for calculating geometrical phase in the context of coastlines, where differences in elevation fall well below 16 m?

- Line 302: what is meant by a “robust” regression? Please clarify
- Lines 335-336: on what basis are the standard deviation values of 3 and 1 mm/yr for ALOS-1 and Sentinel LOS velocities assigned respectively? Please clarify

I give further minor suggestions to word choice and sentence structure as tracked changes in the attached word document (which already incorporate some of the changes suggested above).

REFERENCES

- Saintilan, N., Kovalenko, K. E., Guntenspergen, G., Rogers, K., Lynch, J. C., Cahoon, D. R., Lovelock, C. E., Friess, D. A., Ashe, E., Krauss, K. W., Cormier, N., Spencer, T., Adams, J., Raw, J., Ibanez, C., Scarton, F., Temmerman, S., Meire, P., Maris, T., ... Khan, N. (2022). Constraints on the adjustment of tidal marshes to accelerating sea level rise. In *Science* (Vol. 377, Issue 6605, pp. 523–527). American Association for the Advancement of Science (AAAS).
<https://doi.org/10.1126/science.abo7872>
- Temmink, R. J. M., Lamers, L. P. M., Angelini, C., Bouma, T. J., Fritz, C., van de Koppel, J., Lexmond, R., Rietkerk, M., Silliman, B. R., Joosten, H., & van der Heide, T. (2022). Recovering wetland biogeomorphic feedbacks to restore the world’s biotic carbon hotspots. In *Science* (Vol. 376, Issue 6593). American Association for the Advancement of Science (AAAS).
<https://doi.org/10.1126/science.abn1479>

Reviewer #1 (Remarks to the Author): (Reviewers comments in normal text, response to reviewers are in bold)

Ohenhen et al., provide an interesting and important analysis of VLM rates along the US Atlantic Coast. Their findings regarding wetland vulnerability are particularly notable given the lack of certainty in the community regarding the ecosystem's ability to adapt to sea-level rise. The article is overall well written - minor comments are highlighted in the attached pdf.

We thank the reviewer for their comments.

A bit more severe concern is the general ability of the figures to present relevant information. In figure 1, I do not see the purpose of the mean flood days increase. How these are derived is not discussed. What geophysical processes are included in the 'coastal waters', what a 'local threshold', and even where this is occurring is unclear. My guess is that you are taking a point located at a tide gauge (e.g., Charleston) and then propagating this to say the whole city is experiencing more days in recent years. As this paper highlights the importance of spatial variability, this would not seem to be a consistent message.

Figure 1 is provided as a background to introduce the study area, the InSAR dataset, and the vulnerability of the US Atlantic coast to sea level rise. This vulnerability is emphasized using the increase in the mean flood days at tide gauges across the region. We did not process or perform any analysis using the flood data, this data was obtained from NOAA tide and current data and is referenced in the figure caption. The names attached to each bar chart are the name of the tide gauge data (e.g., Charleston tide gauge, the battery tide gauge), not the cities, we have added a note clarifying this in the figure caption.

Figure 2a is essentially just a wash of yellow. The subsidence panel on the right does little to help. Is all of NC, except for the few red hotspots, subsiding at the same rate? That yellow rate looks anywhere between -1 and -3 mm/yr, which is a significant difference given the range of magnitudes. You need to figure out a different method (e.g., a discrete colorbar) to bring out the features. More importantly, it does not seem to agree with Figure 4b, which should be the same data. Specifically, there is no data in the southwest corner of the Delmarva peninsula in 4b but there does seem to be data there in Fig 2.

We thank the reviewer for this observation. We have modified Figure 2, and supplementary figures 2 and 3 to discrete colorbars.

Figure 4b is the exact subset of figure 2c, with the area extent shown as the rectangle. Note that this map is not the VLM map but the subsidence map (figure 2c), hence any appearance of omitted data in the southwest corner of the Delmarva peninsula is either because the uplift signals are not displayed in the map or due to the large spatial scale of Fig 2c, which magnifies each InSAR pixel. We highlighted this in the title and figure caption.

Figure 3 is flat out missing uncertainties in several places. The units are a) inconsistent with the rest of the paper and b) increase the difficulty of interpretation.

Also, the projection analysis seems tangential to the paper and is poorly discussed. What justification is there to extend 13 InSAR rates 78 years into the future? How do the methodologies of the IPCC VLM and InSAR compare?

We thank the reviewer for this observation. We note that the standard deviation of the VLM rates is estimated for each pixel; thus, it is spatially variable. In locations of some tide gauges, the standard deviation is minimal (e.g., Woods Hole), while in others, it is larger (e.g., Lewes), we have added a note about this in the figure caption. We have modified the units of figure 2 to mm and mm/year to be consistent with the entire paper. Projections of RSLR and the variables, which impact relative sea level rise (such as subsidence) are important in predicting coastal vulnerability, we have modified the title of the paper (consistent with other reviewers' suggestion) to reflect the overarching topic of coastal vulnerability. A synergy of the relative SLR and coastal vulnerability is discussed in the concluding paragraph.

Current IPCC projections of relative RSLR (Garner, et al. 2021; Fox-Kemper et al., 2021) include VLM rates at the location of tide gauges, which are linearly extrapolated through 21st century. To be consistent with IPCC projections, we compared a linear projection of our VLM rates to highlight the underestimation/overestimation of such VLM projections. We have added a few sentences on lines 128-133 about the methodology of derived IPCC VLM rates.

Figure 4: It is unclear how VLM pixels are tied to the land cover classes. Please elaborate on this sentence in the methods:

1. "We combined VLM velocity with the NLC by resampling the VLM velocity for the different NLC."

We thank the reviewer for this observation. We have added a sentence about this on lines 399 – 400.

Figure 5: Why do you correct the TG with GPS instead of the InSAR? This seems to defeat the purpose (and is inconsistent with the method used in Figure 3). Additionally, the nearest GPS to TG can be quite far and unrepresentative of actual VLM.

The comment seems unrelated to figure 5, but the reviewer perhaps meant Figure 3. In figure 3, we are comparing rates of VLM at tide gauges only, not using the SLR record at the tide gauge. In figure 5, we incorporate rates of absolute SLR from tide gauges to calculate the vulnerability of the wetlands. To prevent double counting the VLM rates, we corrected the tide gauges using GPS data. While tide gauges may be far from GPS stations, in some regions, in our analysis, no tide gauge was more than 800 m away from at least 1 GPS station.

Most seriously, I do not see sufficient evidence to trust the generated VLM rates. The underlying LOS dataset is not reproducible. There are myriad community tools for InSAR processing and analysis that are well established, widely vetted, and carefully documented (e.g., ISCE, MintPy). The WabInSAR (Shirzaei, 2013) used has none of those properties and is not available to the community (as far as I am aware). As such, this contradicts Nature's commitment to open science (<https://www.nature.com/nature-portfolio/editorial-policies/reporting-standards>) and falsifies the authors' statement in the `Data and materials availability` section. While the theoretical basis of WabInSAR is documented and shown to perform well for a particular sensor and study area with substantial displacement (Shirzaei, 2013), InSAR algorithm performance varies widely given the number of error sources that vary spatiotemporally. This is especially relevant given the mm-scale displacement (vs centimeter scale in Shirzaei, 2013). However, no discussion of error sources and minimal uncertainty metrics are presented in the submission.

The WabInSAR code is widely vetted and has been used in the processing of InSAR data in a variety of research projects dealing with fast, slow, steady or instantaneous deformation signals, which are published in over 50 peer-reviewed journals such as science, nature, and PNAS and cited more than 1500 times.

The code is open-sourced and has been available to the community for the past 8 years. Here is the download link to the software:

<https://drive.google.com/drive/u/0/folders/1ZXPGPJPY94jlpRAx9-eGezB31K8VVJG5>. Here is a link to the website where the software is hosted:

<https://sites.google.com/vt.edu/eadar-lab/software?authuser=0>. Also, the WabInSAR source code and test dataset are carefully peer-reviewed and publicly available through Shirzaei et al 2019 EPSL

(<https://www.sciencedirect.com/science/article/pii/S0012821X19301736>), see the supplementary materials.

Based on IP tracking, the community has downloaded the software more than 200 times. Furthermore, we have made public the final deformation maps generated by WabInSAR (<https://sites.google.com/vt.edu/eadar-lab/data-sets?authuser=0>), which are downloaded more than 400 times and are used in many research articles and governmental reports. Noteworthy, USGS has just completed an independent review of our codes and products and is planning to include our VLM products in their HERA tool (www.usgs.gov/apps/hera) for public use.

Due to US sanctions (<https://home.treasury.gov/policy-issues/financial-sanctions/sanctions-programs-and-country-information>), the ISCE software (and other JPL/NASA tools) is not available to the citizens of many countries (primarily under-developed) including Iranians. Note that one of the co-authors of this manuscript is of Iranian origin. Thus, we refuse to use it while this discriminatory policy is in place. To our knowledge, the ISCE software has only been used by a small group, mainly at JPL, who provided a mixed review of its performance.

The WabInSAR code can be used with either GMTSAR (Sandwell et al., 2011) a truly open-source and robust code or GAMMA (Werner et al., 2000) a commercial code. Both

of these softwares are free from US sanctions and have been tested and validated thoroughly. For instance, Figure below shows the use-base for GMTSAR.

[redacted]

Figure 1: GMTSAR user-base. Courtesy David Sandwell.

As for MintPy, it is a basic post-processing module that applies weighted least squares to a series of connected interferograms to create a time series of Line-of-sight deformation (Yunjun et al. 2019). MintPy is most suitable for educational purposes and should not be compared with advanced multitemporal InSAR algorithms, such as WabInSAR and STAMPS (Hooper, 2008), that perform sophisticated algorithms for identifying elite pixels, sparse phase unwrapping and correcting environmental artefacts.

Additionally, the Line-of-sight velocities are easily reproducible using the WabInSAR code or any trusted InSAR algorithm. We have included a link to the WabInSAR code with this submission for your review.

We addressed all error sources and uncertainty metrics during processing on lines 300 – 321 including cited references. We have also included additional uncertainty discussion on lines 364 – 395.

Uncertainties are primarily reported regarding the fit to GPS stations. While these numbers look good enough, there are only a relatively small number left out for validation (41/460) permitting overfitting. More importantly, the GPS are primarily constraining the long wavelength signals. In between GPS stations, short wavelength InSAR displacement signals are subject to noise. Indeed, how many GPS stations are in wetlands?

Firstly, our InSAR-GPS combination is not a training-testing model but a joint inverse modeling to resolve 3D displacement field. The ideal 3D displacement field should be able to fit all the GPS data and well as InSAR datasets. Secondly, while we extracted 460 GPS stations, some were outside the SAR frames and were not used in the analysis. We

have now modified the text and figure S7, to show only 173 stations that were close enough to the InSAR pixels. Thirdly, we addressed uncertainty in the context of accuracy and precision. The precision is reported in terms of standard deviation following propagating observation error to the parameters (Mikhail & Ackerman, 1982) as shown in Supplementary Figure 3. To perform a robust accuracy assessment, we choose independent GNSS observations as ground truth. To this end, the central limit theorem in statistics requires a random sample size larger than 33, thus, 41 GNSS stations for validation is statistically justified. Lastly, as discussed throughout the text, we are not extracting InSAR pixels for just wetlands, but the entire coast. Note that GPS stations are not installed in wetlands; if they are, they do not measure the shallow compaction due to the depth of their anchors. Thus GPS measurements would not be compared with that of InSAR measurements on wetlands. See Shirzaei et al. (2021) for additional discussion. Furthermore, to underscore the importance of our study to capture the dynamic processes on wetlands, consider the wetlands in the Blackwater National refuge. This area is recognized as a hotspot for wetland loss in the Chesapeake Bay area (Nerem et al., 1998). Despite, the uplift signal associated with current groundwater recharge in the area (discussed on lines 125 to 126), our observation captures the loss of the wetlands associated with shallow processes (Fig. 2a). This highlights the retention of both the short and long wavelength in our final InSAR VLM map.

This is especially important given the challenges of InSAR in wetlands, which decorrelate when inundated, and are almost by definition in vegetated areas (which are also susceptible to decorrelation). In addition to adding timeseries noise, decorrelation leads to challenges in unwrapping. How are these addressed? What is the temporal coherence of the wetlands? What do some timeseries look like?

Our study is not the first study using InSAR for wetland monitoring as stated on line 85 with cited reference. However, our study is the first study to accomplish this at this spatial-scale. We emphasize again that we did not perform InSAR on wetlands alone, InSAR pixels on wetlands after oversampling constitute 3.7% of the total obtained pixels (1.4 wetland pixels compared to 38 million total pixel; see lines 116 and 436). Perhaps the plotting on figure 5 creates the illusion of an oversaturation of wetlands pixels, we have added a new supplementary Fig. 6 over the Chesapeake Bay area to highlight this point. To prevent decorrelation issue associated with inundation of wetlands, we screened the SAR (Sentinel-1) images, prioritizing images obtained during low tides; with the maximum exposed surface. This helps to maximize InSAR pixels over wetlands and minimize any errors. We have included a sentence about this on lines 291 – 293. We used a temporal coherence threshold of 0.7, selecting elite pixels with very high temporal coherence and discarding noisy pixels, we have added a sentence about this on lines 309 - 310 (Fig. 2 b and c). A timeseries representing the VLM is not obtainable due to the different temporal scales of the combined satellite images, without overlapping periods and that we performed the joint inversion of InSAR and GNSS for rates (see also Blackwell et al., 2020). Here, obtained LOS displacement velocities from the 2 satellites are combined with GNSS observations to obtain the 3D displacement velocity as discussed in lines 336 – 361 using equation (1) to equation (5).

[redacted]

Figure 2: (a) VLM across the Chesapeake Bay area. (b) Temporal coherence map of Sentinel-1 frame over the Chesapeake Bay area. (c) Temporal coherence map highlighting the threshold of 0.7.

Where uncertainties are reported, it's ambiguous as to what they even are. The data file and Figs 3, 5, Table 1, Eq 6, Fig S3, show a standard deviation associated with the VLM rate. Is this the standard deviation of the time series? The standard deviation of bootstrapped rates? A measure of the residue of the linear fit? How does it compare to the rates? The colors makes it impossible to tell.

The uncertainties are standard deviations obtained through propagating the observation errors into the parameters (Mikhail & Ackerman, 1982). We have added a section in the methods section addressing the uncertainties in the observations and have explicitly defined the uncertainties where they are used, e.g. Table 1.

How do localized results compare to existing results (e.g., Bekaert et al., 2017, Scientific Reports)?

The results are consistent with previous localized geodetic studies in the region, with VLM values between -2 to -3 mm/yr. We previously cited the study of Buzzanga et al., 2020 on Hampton road to highlight this, but we have now included Bekaert et al., 2017.

Can you provide the data in a more usable and realistic way? Why are these point measurements if they are 50m spatial resolution? A raster would be more appropriate and much easier to work with.

The 50 m resolution pertains to the original spacing of pixels within an interferogram. While performing advanced Multitemporal InSAR we identify pixels that contain mostly noise and discard them. From this point on, the observations are no longer on a regular grid. Since this dataset contains precisely geocoded spatial data, vector format data is more appropriate. Additionally, a gridded-raster map would undermine the InSAR's high-

fidelity accurate spatial measurements, which is one of its benefits. For ease of handling, we separated the VLM map by states on the east coast and provided a single CSV file of the data available in the data repository.

Given these considerations, I do not think this article warrants publication in Nature Communications.

We thank the reviewer for their comments. We believe that our replies addressed any major concerns regarding the data, uncertainties, and figures.

Additional comments on attached main text PDF:

Line 30: what does significant mean?

A teaser sentence is not required and has been removed.

Line 44-45: This doesn't make sense?

We have modified the paragraph and the sentence.

Line 62: this statement is not supported by figure 1

We have deleted the reference to figure 1.

Line 120: how does what is happening in CA have an impact have on CBay?

We have deleted the reference to California and only referenced the cited work.

Line 159: what are these uncertainties?

Uncertainties are the standard deviation associated with each InSAR VLM. We have defined this in lines 384 – 389.

Lines 193 - 194: this is unjustified. as your InSAR is tied to GPS, the subsidence it reflects is tied to the depth of the GPS benchmark. a thorough analysis of the GPS depths would be required to make this statement. see (Karegar, M. A., Larson, K. M., Kusche, J., & Dixon, T. H. (2020).)

We have reworded this sentence to say InSAR measurement provides information of both shallow and deep processes.

Line 234: please elucidate/reference

We have included two references.

Lines 261 – 263: quantify/define

We have modified this sentence and the entire paragraph.

Line 277: what does this mean? grandiose statements like this should not be used without careful attention to detail.

The concept of tipping points has emerged as a growing research topic in climate research. A tipping point is defined as a critical threshold at which a tiny perturbation can qualitatively alter the state or development of a system. Tipping points have been defined with reference to temperature, Antarctic and Greenland ice sheets and ecosystems. We have included references for this sentence.

Additional comments on attached supplementary document PDF:

Fig. 3: Do not use a divergent colormap for sequential data.

We have modified the figure colormap.

References

Blackwell, E., Shirzaei, M., Ojha, C., & Werth, S. (2020). Tracking california's sinking coast from space: implications for relative sea-level rise. *Science Advances*, 6(31), 4551. <https://doi.org/10.1126/sciadv.aba4551>

Hooper, A. (2008), A multi-temporal InSAR method incorporating both persistent scatterer and small baseline approaches, *Geophys. Res. Lett.*, 35(L16302, 10.1029/2008GL034654.)

Mikhail, E. M., and F. E. Ackermann (1982), *Observations and least squares*, University Press of America.

Nerem, R., van Dam, T., & Schenewerk, M. (1998). Chesapeake bay subsidence monitored as wetlands loss continues. *Eos, Transactions American Geophysical Union*, 79(12), 149–149. <https://doi.org/10.1029/98EO00110>

Sandwell, D., R. Mellors, X. Tong, M. Wei, and P. Wessel (2011), Open radar interferometry software for mapping surface deformation, *Eos Trans. AGU*, 92(28), doi:10.1029/2011EO280002

Sandwell, D., R. Mellors, X. Tong, M. Wei, and P. Wessel (2011), Open radar interferometry software for mapping surface deformation, *Eos Trans. AGU*, 92(28), doi:10.1029/2011EO280002.

Shirzaei, M., Freymueller, J., Törnqvist, T. E., Galloway, D. L., Dura, T., & Minderhoud, P. S. J. (2021). Measuring, modelling, and projecting coastal land subsidence. *Nature Reviews Earth & Environment*, 2(1), 40 – 58. <https://doi.org/10.1038/s43017-020-00115-x>.

Werner, C., U. Wegmüller, T. Strozzi, and A. Wiesmann (2000), Gamma SAR and interferometric processing software, paper presented at Proceedings of the ers-ensat symposium, gothenburg, sweden, Citeseer.

Yunjun, Z., Fattahi, H. and Amelung, F., 2019. Small baseline InSAR time series analysis: Unwrapping error correction and noise reduction. *Computers & Geosciences*, 133, p.104331.

Reviewer #2 (Remarks to the Author): (Reviewers comments in normal text, reponse to reviewers are in bold)

There is an ongoing debate in the literature regarding the relative contribution of subsidence to RSLR, particularly on the US Atlantic coast. In this paper, the authors use SAR, GNSS, and tide gauge data to produce coast-wide measurements of vertical land motion and subsidence. Results provide strong evidence that wetlands on the Atlantic coast are indeed experiencing widespread subsidence, and as a result, coastal wetlands are increasingly vulnerable to inundation. With minor revisions, this will be an exciting contribution to the literature.

We thank this reviewer for their comments and excellent suggestions. We are grateful for your voluntary, anonymous review and have incorporated all suggestions in the manuscript as discussed below.

Line 81: Clarify that although many assessments of coastal vulnerability do not account for both shallow and deep subsidence, the studies you cite here are ones that DO. May want to mention a few studies that do not account for shallow subsidence.

We have included one example of such study on line 81.

Line 99: What is the date range for the GNSS observations?

The GNSS observations are from 2007 – 2020. This has been included on line 99. Thank you for the observation.

Line 128: The subsidence rate in NYC appears to be ~1.5 mm/yr. This is certainly not something to ignore, but it doesn't jump out at me as a hotspot of subsidence. Boston appears to be subsiding faster (~3 mm/yr). Or maybe you are simply listing large Atlantic coast cities, most of which are undergoing subsidence?

Boston has been included in the cited examples of subsiding cities on line 118.

Lines 129-130: I do not see that subsidence is high in the Chesapeake Bay region as a whole. It appears that areas around Delaware Bay and the eastern and southern Delmarva Peninsula are subsiding rapidly, but much of the area around the Chesapeake Bay has little subsidence or is uplifting (as you note later in this paragraph).

Yes, this is correct. We have modified this sentence in this paragraph to reflect the subsidence is in some areas.

Lines 137-138: Are the IPCC's VLM rates based on tide gauge data?

Yes, the current IPCC VLM rates were obtained from tide gauges, we have included a sentence about the measurement of IPCC VLM on lines 126 – 130.

Lines 140-144: What is your buffer for an “accurate” measurement of VLM (not an over or underestimate)?

We calculated the percent difference of InSAR VLM rate with respect to IPCC VLM. This has been clarified on lines 141 – 143, included in Figure 3 and supplementary table 2.

Line 149: How are you defining “exposure” to subsidence?

We defined exposure here as the measure of the dominant land cover types. Included on lines 152 – 153.

Lines 153-156: This sentence implies that some land cover types experience subsidence while others do not, but these four categories account for 96% of all land cover types in the study area. I suggest simply stating the four categories and corresponding percent land cover (i.e., cut “Subsidence exposure follows dominant land cover, where”.

Thanks for the suggestion. This sentence has been modified on lines 155 – 156.

Lines 160-161: Yikes, 11.8 mm/yr of subsidence. Where is this?

Close to Beaverdam branch in Delaware. Coordinates→ lat: 38.5878, lon: -75.4443

Line 172: By “agrarian land”, do you mean cultivated crops plus pasture/hay?

Yes, included in line 174

Line 192: As you mention below, VLM as measured by InSAR captures all subsidence processes (shallow and deep). Tide gauges, however, typically do not record shallow subsidence. Thus, differencing these two types of data will give you an overestimate of VR (because shallow subsidence is not subtracted out). The inclusion of shallow subsidence in SLR rates would likely indicate that more regions are experiencing a VR deficit, or a greater deficit. Please make note of this in the text.

By accounting for both shallow and deep subsidence using InSAR measurements and removing any subsidence signal from the SLR data using GNSS, we are assessing the vulnerability of the wetlands to both the shallow and deep processes, which is in line with the recent study by Saintilan et al. (2022) who demonstrate through field observation that subsidence in the shallow substrate increases the vulnerability of wetlands. We note that various studies of basin sediment compaction indicate that the settlement of sediment under the weight of the added load can continue for decades to centuries (Kooi & Vries, 1998; Meckel et al., 2006; Törnqvist et al., 2008). Thus a realistic VR analysis for the time scale of 21st century considered here is required.

Line 200 (and methods): Typically, GNSS do not measure shallow subsidence either, so they do not solve the overestimation of VR mentioned above.

See the response above.

Lines 211-213: Very clear summary of results. Note that vulnerability values would be even greater if shallow subsidence measurements were included in SLR!

Thank you for your nice comments. Yes, that is the case.

Lines 216-224: Nice summary of the importance of this work.

Thank you for your nice comments.

Lines 238-240: These two sentences are duplicated in the results. Cut from one of these sections.

Thank you for the observation. We have modified the two sentences.

Figure 1: Excellent figure. Packs in a lot of information without feeling cluttered or confusing. Very nice. Note that the study area is the US Atlantic coast from New Hampshire to Florida (excluding Maine), so not quite the whole US coast.

Thank you for your nice comments. We have noted that the study area is from New Hampshire to Florida in the figure caption.

Figure 2: How are panels a and c different? Is c just the negative (subsidence) component of a? That is fine, but clarify in the caption.

Yes, Fig. 2c is the negative component of Fig. 2a, we have modified the caption.

Figure 3: Please make all text larger! Is the blue shaded region (2 SD) present in each panel but too narrow to see in some cases (Woods Hole, Bridgeport, etc.)? If so, note this in the caption.

We have increased the texts in the figure and have noted the narrow standard deviations in the figure caption.

Supplemental Figure 5: The blue triangle corresponds to the red curve and the red triangle corresponds to the blue curve? Please fix.

This was an error in the figure caption. It has been fixed.

References

Kooi, H., and J. De Vries (1998), Land subsidence and hydrodynamic compaction of sedimentary basins, *Hydrology and Earth System Sciences Discussions*, 2(2/3), 159-171.

Meckel, T. A., U. S. ten Brink, and S. J. Williams (2006), Current subsidence rates due to compaction of Holocene sediments in southern Louisiana, Geophysical Research Letters, 33(11), doi:10.1029/2006gl026300.

Saintilan, N., et al. (2022). Constraints on the adjustment of tidal marshes to accelerating sea level rise. In Science (Vol. 377, Issue 6605, pp. 523–527). American Association for the Advancement of Science (AAAS). <https://doi.org/10.1126/science.abo7872>

Törnqvist, T. E., D. J. Wallace, J. E. Storms, J. Wallinga, R. L. Van Dam, M. Blaauw, M. S. Derksen, C. J. Klerks, C. Meijneken, and E. M. Snijders (2008), Mississippi Delta subsidence primarily caused by compaction of Holocene strata, Nature Geoscience, 1(3), 173.

Reviewer #3 (Remarks to the Author): (Reviewers comments in normal text, reponse to reviewers are in bold)

ARTICLE SUMMARY

Vertical land motion (subsidence vs uplift) remains poorly monitored globally, despite being necessary to calculate relative sea level change at the coast accurately. This is particularly worrying, given the dual threat of accelerating eustatic sea level rise and subsidence (often linked to anthropogenic activity) that is increasingly inundating coastal cities, forests, agricultural land, and wetlands. To address this important knowledge gap, the authors used satellite-derived data to map vertical land motion rates across the US Atlantic coast, and show that most of the coastline is subsiding. For saltmarshes in particular, the authors estimate that between 58 - 100% of marshes are losing elevation upon calculating the difference between local sea level rise and vertical land motion rates. The authors point to an even greater risk social, economic, and ecological systems faced by the US Atlantic coast now that subsidence has been mapped for the first time. Moreover, vertical land motion rates used by the IPCC differ from the author's calculations, indicating greater uncertainty in current assessments of future coastal flood risk. The authors advocate for the need to better account for subsidence when planning coastal adaptation strategies.

GENERAL COMMENTS

Including vertical land motion in assessing coastal flood risk remains an important research gap, especially in the context of saltmarshes where a small difference in relative sea level rise rate can define whether they survive or drown. If up to 100% of marshes across the US Atlantic coast are indeed at risk of drowning due to subsidence, this represents an existential crisis for saltmarshes – especially if the US Atlantic coast situation is similar to other parts of the globe. This work overturns the current paradigm that marshes are expected to survive sea level rise due to a positive feedback between increased accretion and prolonged inundation by sea level rise. Moreover, the manuscript compliments the recent publication of Saintilan et al. (2022), who demonstrate from field observations that less than half of the sediment accreted on marshes is translated to surface elevation gain (made even worse by sea level rise, as the addition of more material increases shallow compaction). I therefore believe this manuscript is novel, timely, and of relevance to the wide readership of Nature Communications.

We appreciate this reviewer's time and efforts in evaluating the manuscript. We deeply appreciate all of your insightful comments and suggestions, which have been integrated into the manuscript.

I would like to see the authors:

- State more explicitly in the text what the uncertainty in the vertical land motion rates (derived from InSAR data) are, how these were assessed, and what these mean for confidence in the reported subsidence rates. These points are covered throughout the text, but I believe they should be made easier for the reader to understand. For example, standard deviation is used as the uncertainty metric, and its derivation is described in lines 110-112 (for east and north velocities) and lines 133-134, 348-357 (from comparing InSAR upwards velocities to the GNSS

data, also reported in Fig. 2). I'm unclear if both feature in the overall accuracy assessment. Moreover, do these measures account for the accuracy of the SAR sensors?

We have moved all discussions on uncertainties and the standard deviation, which were presented in bits throughout the text to a coherent subheading in the methods section titled "InSAR Validation Analysis and Reported Uncertainties." This is on lines 359 – 385. Here we explain the metrics for validation, how they were obtained and the uncertainties/standard deviations reported in the manuscript. On lines 106 – 108 of the manuscript (introduction of the InSAR VLM) and all first occurrences of reported uncertainties (lines 140 and 164) regarding the NLCs and VLM projections, we reference the method section where the uncertainties are discussed.

- Relate the relevance of their findings to other coastal environments beyond the US Atlantic coast – especially in the context of river deltas where subsidence is a key threat and often the result of development of the river catchment resulting in reduced freshwater flow and sediment flux to the coast.

We thank the reviewer for this insightful suggestion. In the final paragraph, we have expanded our discussion to incorporate other coastal environments and discuss the threat of subsidence to cities found in river deltas that are subsiding rapidly.

The quality of writing is high, manuscript is formatted correctly, and the figures and supplementary materials are helpful and well presented. I believe the manuscript is robust and well referenced. Some points could be clearer, and I highlight these below.

We thank you for your comments and suggestions, we have used the review to clarify the unclear points in the manuscript, as itemized below.

SPECIFIC COMMENTS

Please refer to line numbers in the PDF version of this manuscript:

- Line 1: May I suggest two changes to the title: (1) "Increased" implies vertical land motion has recently begun to affect US Atlantic marshes, which is not the case. It's that vertical land motion has not been properly accounted for until now. "Hidden" or "Overlooked" may be more appropriate; (2) Since the effect of vertical land motion on forests, agricultural land, and cities are also being considered, I recommend removing "wetlands" from the title.

Thank you for this excellent suggestion. We have modified our title according to your suggestion.

- Line 17: Given that cities are being considered too, is it better to replace "ecosystems" with "environments"?

We have modified this word to environments.

- Line 18: remove "heterogeneous tidal ranges and land cover types" as the focus is on subsidence

Removed.

- Line 20: since GNSS are also used in the analysis, maybe “satellite data” could be used instead of “synthetic aperture radar” as a catch-all term

Modified to satellite data. Thanks for the suggestion.

- Line 24: The focus on coastal marshes in the sentence starting “We estimate...” leaves the reader wondering why wetlands forests / agricultural areas / developed regions aren’t summarised. I would add a connecting sentence to emphasise why wetlands become a key point

We have included a preceding statement on lines 23 – 24 stating that wetlands are the dominant land cover in the region.

- Line 30: Is subsidence a climate-related risk? I would argue it is largely anthropogenically-induced

We have removed the teaser sentence, since it is not required.

- Lines 41-42: “flooding” and “storm surge” are identified as separate hazards, though the former is often a consequence of the latter. Perhaps specify “river flooding” instead.

We have modified the introductory paragraph and removed the occurrence of flooding and storm surges.

- Lines 45-47: For the example given here, I don’t think the link between coastal ecosystems and coastal cities is particularly strong. I suspect many of the megacities function without much reliance on local coastal ecosystem services

We have modified the introductory paragraph.

- Line 47: “Coastal wetlands, which shelter most coastlines” is a bit of a sweeping statement. Coastal wetlands tend to occur in sheltered coastal settings

We have modified this sentence in line with the modification to the introductory paragraph to: “Toward the sea, most coastlines are sheltered by coastal wetland ecosystem.”

- Line 47: The statement that “coastal wetlands ... store more carbon ...” is no longer correct according to Temmink et al. (2022). Tropical peatlands hold store more carbon. Sequestration rates are generally higher in marshes, however. I would update reference 8 (McLeod, 2011) with Temmink et al. (2022).

Noted! Temmink et al. (2022) have been included as a reference.

- Line 51: “carbon sequestration” has been repeated from the previous sentence, so should be removed.

We have modified the introductory paragraph.

- Line 54: “(ref .14)” is used rather than the superscript form. These cases should be checked throughout the manuscript.

Ref. is used in place of the subscript form if a number is the last sentence before the citation to avoid conflating the citation with a number exponential or power form. Other cases are when an example is to be cited. We have checked and this is in line with the citation format of *nature communications*. In other cases (lines 65 and 66), we have removed the ref. and used the correct citation format.

- Line 60: “...which threatens coastal ecosystems”. Please summarise how subsidence could threaten coastal ecosystems

Included in lines 58 – 62.

- Line 81: in addition to refs 37-40, Saintilan et al. (2022) should be referenced also

Reference has been included.

- Line 82: “InSAR overcomes these challenges” – please summarise how. E.g., “...by providing high spatial and temporal resolution on elevation change”

Yes, included in lines 84 – 85.

- Lines 110-111: “41 GNSS stations, which were not used in the analysis” – please explain why

We did not use the 41 GNSS stations in the analysis to provide an independent unbiased validation metric. Included in the uncertainty discussion on lines 362 – 288.

- Line 130: I am unclear what is meant by “bounds” when discussing uplift in the Chesapeake Bay area

This means areas surrounding the Chesapeake bay. It was not very clear and we have modified this in lines 123 – 125.

- Line 136: By “these measurements”, do the authors mean the rate of sea level change? Please clarify

Yes, we mean sea level change. We have modified this on line 129.

- Line 137: It would be helpful to know how the IPCC have projected rates of vertical land motion. Do they also use SAR or a different method?

The IPCC projected VLM using tide gauge data, we have included a sentence about this on lines 129 – 134.

- Lines 149-150: The implication here is that subsidence rates are embedded in the NLCs, which is incorrect. Please clarify

Clarified on lines 153 – 155.

- Lines 153-154: “Subsidence exposure follows dominant land cover” – I’m unsure what is meant here. Please clarify

We have clarified this in lines 155 – 156.

- Line 173: “This loss of agricultural land [due to subsidence]” – please clarify this somewhat. The land would be lost to salinisation, which increases due to increased coastal flooding because of subsidence, correct?

This sentence has been removed and is now on line 242 – 243, where the reason for the loss of agricultural lands are discussed.

- Line 184: “a modification of” – since the modification isn’t immediately explained, the reader is left wondering. I would remove this and allow the text later to explain the modification after VR is explained in the next sentence

Removed.

- Line 187: The cut-off of 0.5 mm/yr chosen is not adequately explained. Please state why this cut-off was used.

This threshold was necessary to account for uncertainties in the measurements, the dependence of sea level rates on the recording period, and the potential for short-term marsh accretion rates to fluctuate with historical SLR rates. We have included a reference for this and referenced the reader to the methodology where this is explained.

- Line 189: is it that marshes are “vulnerable to SLR”, or are they vulnerable to the effects of sea level rise (i.e. excessive inundation that exceeds the physiological tolerance of marsh plants)?

The vulnerability of marshes is defined based on ‘excessive inundation that exceeds the physiological tolerance of marsh plants.’ Here, we are referencing just the loss of elevation with respect to SLR. On lines 255 – 256, we acknowledge this shift in accretionary balance may not mean the marsh is completely lost due to the ecological shift towards flood-tolerant plants.

- Line 206: “with little dependence on existing marsh elevation” is unclear – please clarify

This is in reference to either low-elevation or high-elevation marshes and is now clarified using parentheses.

- Line 234: “the oxidation of organic materials” – please provide a reference

References have been included on line 238

- Line 254: “...but do not account for future accelerations in...” - do the authors mean their work does not account for changes in the rate of SLR? Please clarify.

We are referring to the difference in the rates of SLR under various RCP scenarios. Clarified on lines 266 – 267.

- Lines 259-261: This sentence could be clearer. I believe the authors are saying that accounting for VLM is needed to predict the true risk of coastal flooding.

This has been modified on lines 261 – 262.

- Line 273: “vulnerability estimates” – perhaps this is too generic. Do the authors specifically mean vertical land motion rates?

We have modified this to “high-resolution estimates of VLM” on line 275.

- Line 296: The SRTM has a vertical accuracy of +/- 16 m. Does this reduce its effectiveness for calculating geometrical phase in the context of coastlines, where differences in elevation fall well below 16 m?

No. The effect of DEM error on the interferometric phase is a function of the perpendicular baseline. For the case of Sentinel-1, the baselines are less than 100 m, thus a SRTM DEM error results in an error less than a fraction of mm. We further correct each interferogram for the effect of “DEM residual” error, which is a routine procedure in advanced multitemporal InSAR analysis.

- Line 302: what is meant by a “robust” regression? Please clarify

This is based on the study by Shirzaei & Walter (2011). During a robust regression, a function fits a dataset using a reweighted least squares approach, and the observation weight is updated through iterations. The outcome of a robust regression mimics the statistical properties of an L1-norm minimization, which is less sensitive to outliers in the observations (Holland & Welsch, 1977). A citation for the robust regression has been included in the text on line 311.

• Lines 335-336: on what basis are the standard deviation values of 3 and 1 mm/yr for ALOS-1 and Sentinel LOS velocities assigned respectively? Please clarify

This is based on previous studies performed by Blackwell et al., 2020 and Ohja et al., 2018, we have included a statement about this on lines 344.

I give further minor suggestions to word choice and sentence structure as tracked changes in the attached word document (which already incorporate some of the changes suggested above).

We thank you again for the excellent reviews. All changes in the attached document were incorporated.

REFERENCES

- Saintilan, N., Kovalenko, K. E., Guntenspergen, G., Rogers, K., Lynch, J. C., Cahoon, D. R., Lovelock, C. E., Friess, D. A., Ashe, E., Krauss, K. W., Cormier, N., Spencer, T., Adams, J., Raw, J., Ibanez, C., Scarton, F., Temmerman, S., Meire, P., Maris, T., ... Khan, N. (2022). Constraints on the adjustment of tidal marshes to accelerating sea level rise. In *Science* (Vol. 377, Issue 6605, pp. 523–527). American Association for the Advancement of Science (AAAS). <https://doi.org/10.1126/science.abo7872>
- Temmink, R. J. M., Lamers, L. P. M., Angelini, C., Bouma, T. J., Fritz, C., van de Koppel, J., Lexmond, R., Rietkerk, M., Silliman, B. R., Joosten, H., & van der Heide, T. (2022). Recovering wetland biogeomorphic feedbacks to restore the world's biotic carbon hotspots. In *Science* (Vol. 376, Issue 6593). American Association for the Advancement of Science (AAAS). <https://doi.org/10.1126/science.abn1479>

References

Holland, P. W., & R. E. Welsch (1977), Robust Regression Using Iteratively Reweighted Least-Squares, *Communications in Statistics: Theory and Methods*, A6, 813-827.

Shirzaei, M., & Walter, T. R. (2011). Estimating the effect of satellite orbital error using wavelet-based robust regression applied to insar deformation data. *IEEE Transactions on Geoscience and Remote Sensing*, 49(11), 4600–4605.

Blackwell, E., Shirzaei, M., Ojha, C., & Werth, S. (2020). Tracking california's sinking coast from space: implications for relative sea-level rise. *Science Advances*, 6(31), 4551. <https://doi.org/10.1126/sciadv.aba4551>

Ojha, C., Shirzaei, M., Werth, S., Argus, D. F., & Farr, T. G. (2018). Sustained groundwater loss in california's central valley exacerbated by intense drought periods. *Water Resources Research*, 54(7), 4449–4460.

REVIEWER COMMENTS

Reviewer #1 (Remarks to the Author):

Thank you for the thoughtful and detailed responses to my comments. Many of them are addressed. Here are a few more specific comments and general thoughts.

- In Figure 1, the flood days is never mentioned in the text, and so should not be included. Also, the reference needs updating now.

- In Figure 5 you prevent double counting by correcting using GPS VLM rather than correcting InSAR VLM, which seems strange given that you can presumably get much closer than 800 m (which can be a lot) with InSAR.

- Much of my concern regarding the VLM is alleviated given that the code is available to the community. Thanks for providing the link in the code availability which would have been very helpful at the outset, as a google scholar search for WabInSAR does not reveal it. Additionally, citations to Shirzaei 13 from Google Scholar number 64, which are predominantly focused on tectonics and caused my original concern. That is unfortunate about ISCE and I agree that such discriminatory practices should end immediately. The comments regarding MintPy seem strange given Occam's razor.

- You variously claim the dataset is continuous vs semi-continuous vs a point measurement (vector dataset). Text should be updated to be consistent with the latter point.

- Line 132: The IPCC VLM calculation is a residual term estimated after removing sea-level change due to ocean components and as such would include shallow strata, I think (see Kopp).

- Line 273 states 'Current projections of relative SLR underestimate the contribution of VLM (Fig 3)'. Given shown uncertainties, 5/12 stations are underestimated, which does not support this claim (and the numbers in the paragraph starting on line 128). It seems like 10% difference is used as a significance threshold rather than the actual trend uncertainties. This seems arbitrary and unnecessary.

- It remains unclear to me why this comparison is in here. Results are focused on 2007 - 2020 and don't use projections. Inundation models are introduced at the end but without proper context (273). A small subset of tide gauges, and only one coastline, is used to make dramatic claims about global IPCC projections (Line 273). A more rigorous assessment must be made to support such claims. For example, why a 200 m radius around the tide gauge, and what impact does varying this radius have?

- In the paragraph starting on 246 there seems to be confusion/comingling of forest and forested wetlands

- Tipping points defined in Lenton et al. make no reference to the coast or wetlands and are used in the context of the Earth System. Much work has gone into defining them, and the severity of crossing them is increasingly being eroded by misuse of the phrase as a 'buzzword' to attract attention. A study that applies their formal mathematical definition to potential wetland loss to define a tipping point would be an interesting study, but is not done here.

- The uncertainty quantification is clearer now thanks. Why is the weight of the distance reduced by a factor of 10 in Eq 5?

I remain at a loss why you assign a rate uncertainty for ALOS and Sentinel-1 to the East Coast based on studies in California. Although a time series of VLM isn't obtainable as you say, you have a timeseries for each sensor in LOS from which you could characterize the temporal uncertainty on a

pixel basis. The uncertainty of the time-series will vary with landcover type. This is no small point, as the rates, particularly for the study area as a whole, are barely significant at 1sigma. Potentially important uncertainties remain unaccounted for, including the effect of the GPS record length (unlikely that all 100+ cover the full time 2007-2020), the choice of GPS for the inversion (there's a coastal bias in the validation data set), and nonlinearity from anthropogenic activity (there have been large changes in groundwater extraction and potentially VLM in Chesapeake bay over the last few decades). With regards to the RSL, the uncertainty of interpolating the tide gauge rate to the wetlands is unaccounted for. As you correctly say on lines 270-272, "Local rates of relative SLR often vary vastly from regional SLR." I would expect potentially dramatic differences between open ocean tide gauges and wetlands further inland or separated by e.g., a barrier island. I also don't know how to interpret the vulnerability bounds given that wetland VLM is not significantly different than 0 at 2 sigma.

- I'm still unconvinced of the rates at wetlands. The 'screening' (unclear what this means) to isolate low-tide images sounds important. But the rebuttal comment "Our study is not the first study using InSAR for wetland monitoring as stated on line 85 with cited reference " is problematic. Mapping and monitoring are very different endeavors. More importantly, none of the 5 studies cited on line 85 are measuring vertical land motion in wetlands with InSAR. They are primarily focused on water level changes in wetlands. Although I am not an expert on either InSAR or wetlands, none of my searches on google scholar show analyses of VLM from InSAR in wetlands.

Ohenhen et al., have clearly identified an important shortcoming with respect to the lack of VLM in understanding vulnerability to sea-level hazards and developed a novel dataset for addressing it. Their results are suggestive of threats to coastal agriculture and wetlands. However, it would seem prudent to first carefully evaluate the ability of InSAR to extract VLM in coastal wetlands and agricultural lands, through comparisons with in-situ measurements, and to properly quantify the uncertainties of these particularly challenging landcover types in small areas before making large generalizations. That is, to err on the side of caution and attention to detail.

Reviewer #3 (Remarks to the Author):

I am pleased to see the authors agreed with the suggested changes, and believe the manuscript is stronger for it. I greatly appreciate that a clear description of the derivation and use of all error terms now appears in the methods section, and that the final paragraph of the discussion now emphasises why measuring vertical land motion in large river deltas is important.

I believe the authors have fairly reviewed all suggestions, and justified all amendments appropriately. I have no further comments and would be happy to see this manuscript published.

Reviewer #1 (Remarks to the Author): (Reviewers comments in normal text, reponse to reviewers are in bold)

Thank you for the thoughtful and detailed responses to my comments. Many of them are addressed. Here are a few more specific comments and general thoughts.

We thank this reviewer for their comments and suggestions. We have addressed the comments and general thoughts as discussed below.

- In Figure 1, the flood days is never mentioned in the text, and so should not be included. Also, the reference needs updating now.

The flood days is cited in text line 68 and 69, while discussing how RSLR has led to an increase in the flooding during the last decade. We have additionally cited this on line 235. This figure is necessary as a background figure to highlight the hazard to the US Atlantic coast.

- In Figure 5 you prevent double counting by correcting using GPS VLM rather than correcting InSAR VLM, which seems strange given that you can presumably get much closer than 800 m (which can be a lot) with InSAR.

That is correct. InSAR VLM contains both shallow and deep subsidence, while GPS and tide gage measurements only contain deep processes due to anchoring issues. If we correct the tide gage using INSAR VLM, we would incorporate additional measurement errors. What this approach lacks in resolution is accounted for in accuracy.

- Much of my concern regarding the VLM is alleviated given that the code is available to the community. Thanks for providing the link in the code availability which would have been very helpful at the outset, as a google scholar search for WabInSAR does not reveal it. Additionally, citations to Shirzaei 13 from Google Scholar number 64, which are predominantly focused on tectonics and caused my original concern. That is unfortunate about ISCE and I agree that such discriminatory practices should end immediately.

The comments regarding MintPy seem strange given Occam's razor.

We thank you for your comments and suggestions about the code. We are happy to hear that we have addressed some of the reviewer's concern. While the simplicity of MintPy is good for post-processing applications, it is not suitable for our purposes. While MintPy publications fail to provide a citation to the original work by Stefania Usai (2000), she is the first to implement a least square framework for InSAR time series analysis nearly 22 years ago. Since then, the InSAR time series algorithms have evolved significantly and now they implement algorithms for identifying persistent scatterers, atmospheric delay correction, and sparse 3D phase unwrapping. MintPy does not benefit from any of these advancements and is only an implementation of Usai's original work in Python.

- You variously claim the dataset is continuous vs semi-continuous vs a point measurement (vector dataset). Text should be updated to be consistent with the latter point.

In the manuscript, we refer to InSAR measurement as semi-continuous – a high-resolution dataset compared to GPS measurements or accretion data, which we refer to as point measurement in the manuscript. We have modified all references to the InSAR measurements from continuous to semi-continuous for consistency (lines 91, 98, and 228). We have also searched through the manuscript and no reference to the InSAR measurements as a point or continuous measurement are included.

- Line 132: The IPCC VLM calculation is a residual term estimated after removing sea-level change due to ocean components and as such would include shallow strata, I think (see Kopp).

The IPCC projects RSLR estimates using historical measurements at tide gages. These projections consider the contributions to future relative sea levels from steric effects (ocean steric and ocean dynamic effects), ice sheets (Antarctic ice sheets (AIS) and Greenland ice sheets (GIS)), land water storage, glacier and ice cap surface mass balance, thermal expansion, and VLM (Fox-Kemper, et al. 2021). VLM in these projections are estimated from tide gages, as stated on lines 131 – 133. Since these measurements are estimated from tide gages, they do not account for shallow subsidence. This is emphasized by the authors in Fox-Kemper, et al. (2021).

- Line 273 states 'Current projections of relative SLR underestimate the contribution of VLM (Fig 3)'. Given shown uncertainties, 5/12 stations are underestimated, which does not support this claim (and the numbers in the paragraph starting on line 128). It seems like 10% difference is used as a significance threshold rather than the actual trend uncertainties. This seems arbitrary and unnecessary.

We have modified this statement on line 272 to state both underestimation and overestimation (line 273). Indeed both the underestimation and overestimation affect the accuracy of inundation models. The % difference is calculated from the trend of the IPCC and InSAR VLM and their uncertainties. Our consideration for a 10% threshold is to account for a “quantitative” comparison of VLM estimate rather than the “qualitative” estimate of VLM uncertainty overlap. This allows for robustness in the comparison and highlights the error in these estimates.

- It remains unclear to me why this comparison is in here. Results are focused on 2007 - 2020 and don't use projections. Inundation models are introduced at the end but without proper context (273). A small subset of tide gauges, and only one coastline, is used to make dramatic claims about global IPCC projections (Line 273). A more rigorous assessment must be made to support such claims. For example, why a 200 m radius around the tide gauge, and what impact does varying this radius have?

VLM is one of the critical contributions to RSLR used in current projections. However, in the current projections, IPCC VLM already incorporates projections for VLM from tide gages, which are not contemporary rates of VLM. Here, we project contemporary VLM rates from 2007 – 2020 for the US Atlantic coast which can be incorporated into current RSLR projections for use in inundation models. This need for a refinement of VLM is not

novel, by the authors of the IPCC projection estimate's own admission, "In many regions, higher fidelity projections (of VLM) would require more detailed regional analysis" (Fox-Kemper, et al. 2021). We performed such regional analysis in this study and emphasized the need for more rigorous assessment worldwide.

Varying the radius would have a negligible impact! However, since the InSAR resolution is 50 m, we used 200 m to obtain the final VLM estimate using at least ~4 InSAR points to account for any spatial uncertainties and reduce localized high VLM rates. We have added a sentence about this on lines 138 – 139.

- In the paragraph starting on 246 there seems to be confusion/comingling of forest and forested wetlands

We have modified this paragraph to only forests or to coastal forests (line 249).

- Tipping points defined in Lenton et al. make no reference to the coast or wetlands and are used in the context of the Earth System. Much work has gone into defining them, and the severity of crossing them is increasingly being eroded by misuse of the phrase as a 'buzzword' to attract attention. A study that applies their formal mathematical definition to potential wetland loss to define a tipping point would be an interesting study, but is not done here.

This sentence has been modified on lines 267 – 268.

- The uncertainty quantification is clearer now thanks. Why is the weight of the distance reduced by a factor of 10 in Eq 5?

In Eq 5, the numerator is in km, and the average distance between stations is about 10 km. Empirically, we choose the uncertainty to increase with distance by a factor of 1/10. We have tested other factors and found that a factor of 10 is optimal.

I remain at a loss why you assign a rate uncertainty for ALOS and Sentinel-1 to the East Coast based on studies in California. Although a time series of VLM isn't obtainable as you say, you have a timeseries for each sensor in LOS from which you could characterize the temporal uncertainty on a pixel basis. The uncertainty of the time-series will vary with landcover type.

This is no small point, as the rates, particularly for the study area as a whole, are barely significant at 1sigma. Potentially important uncertainties remain unaccounted for, including the effect of the GPS record length (unlikely that all 100+ cover the full time 2007-2020), the choice of GPS for the inversion (there's a coastal bias in the validation data set), and nonlinearity from anthropogenic activity (there have been large changes in groundwater extraction and potentially VLM in Chesapeake bay over the last few decades). With regards to the RSL, the uncertainty of interpolating the tide gauge rate to the wetlands is unaccounted for. As you correctly say on lines 270-272 , "Local rates of relative SLR often vary vastly from regional SLR." I would expect potentially dramatic differences between open ocean tide gauges and wetlands further inland or separated by e.g., a barrier island.

Please note that the WabInSAR code also outputs the LOS time series and velocity standard deviations. Figure 1 shows a histogram of the LOS velocity standard deviation

for a Sentinel-1 frame that covers New York city and its surroundings. As seen, the standard deviations are too small. This is a well-known issue with InSAR and GPS time series analysis codes that overestimate the precision of deformation fields. To mitigate the effect of the overestimated precision, we assigned an uncertainty of one order of magnitude larger than the estimated standard deviation. We found that this selection works well in California and that for the East and West coasts, the uncertainties for SAR satellites are comparable. This justifies the choice of uncertainties for ALOS and Sentinel-1 datasets.

Figure 1. Histogram of the LOS rate standard deviation obtained from multitemporal processing of 183 SAR images in Ascending Frame 130 and Path 33 of Sentinel-1 satellites during 2015/3/12 and 2022/08/26.

We have addressed and acknowledged all uncertainties in our analysis in the manuscript. We included error bars and standard deviations for the values discussed. The wetland vulnerability used a 90% confidence interval to account for all uncertainties. At a lower bound of “2 SIGMA ERROR” wetlands show a 54% vulnerability – THIS IS SIGNIFICANT!

Note that the rate of VLM over wetlands changes very gradually over time and is not affected by anthropogenic processes such as groundwater pumping since groundwater is rarely pumped from aquifers underneath wetlands and the primary driver of elevation change is the addition of new sediments and compaction of sediments under their weight. For example, Meckel et al. (2006) show that in southern Louisiana, sediment compaction rates have been steady for over a century.

For the GPS measurements, “ALL” stations used in our analysis contain only measurements from 2007 to 2020. To ensure robustness, we used stringent criteria in GPS station selection, any GPS station with less than 50 measurements within a single year (i.e. any year between 2007 to 2020) was tossed and not used in the analysis. We have included this in 326 – 327. For the validation dataset, the selection was randomized. Figure 2 below shows a comparison using another set of 44 random GNSS stations selected from the tie GNSS stations. The comparison shows a standard deviation of 0.13 mm/year. To estimate the GNSS rate, we considered the nonlinearity in the GNSS dataset by considering semi-annual, annual, and bi-annual terms in addition to the long-term rates.

Figure 2. Histogram comparing GNSS vertical rates with estimated VLM rates.

For the sea level rise measurements, we did not interpolate relative sea level rise but interpolated the absolute sea level rise. “Local rates of relative SLR often vary vastly from regional SLR” because relative sea level rise contains VLM, while absolute sea level rise do not. The sea level rise data interpolated here has been corrected to remove the VLM (lines 205 – 206). The spatial variability in the rate at which sea level is rising in open sea or further inland are likely to be minimal within similar locations. If we account for VLM, however, this variability increases. Our methodology of absolute SLR not RSLR greatly reduces the uncertainty in wetland vulnerability estimates, this is a major strength in our analysis compared to previous studies, as highlighted on lines 201 – 209.

I also don't know how to interpret the vulnerability bounds given that wetland VLM is not significantly different than 0 at 2 sigma.

The interpretation is this – even accounting for uncertainties in the measurements using a 90% confidence interval does not negate this absolute fact – the US Atlantic coast is vulnerable to SLR. In other words, the US Atlantic coast vulnerability estimates are significant at 90% confidence level. This is an important point we emphasize in this study. For wetlands, the vulnerability is 54% at a lower bound and 100% at a high bound.

- I'm still unconvinced of the rates at wetlands. The 'screening' (unclear what this means) to isolate low-tide images sounds important. But the rebuttal comment "Our study is not the first study using InSAR for wetland monitoring as stated on line 85 with cited reference " is problematic. Mapping and monitoring are very different endeavors. More importantly, none of the 5 studies cited on line 85 are measuring vertical land motion in wetlands with InSAR. They are primarily focused on water level changes in wetlands. Although I am not an expert on either InSAR or wetlands, none of my searches on google scholar show analyses of VLM from InSAR in wetlands.

This work was designed, performed, analyzed, and supervised by experts! Prof. Manoochehr Shirzaei and Dr. Chandra Ohja have more than 10 years of experience and published over 80 peer-review articles on InSAR. Prof. Matt Kirwan has over 50 peer-reviewed publications on wetland monitoring using various techniques and estimating their vulnerability.

As addressed in the previous revision, we did not obtain pixels on every wetland area, but only a few locations, there is no overabundance of InSAR pixels on wetlands. To clarify this point, below is an illustration that shows how SAR backscattering is affected by soil wetness.

[redacted]

Figure 3. SAR backscatters intensity for a different level of wetness. Courtesy Howard Zebker.

When land is wet but not submerged, in fact, the returned radar signal is stronger, and when the land is entirely flooded, there are no returned echoes. Wetlands, by definition, are “where water covers the soil, or is present either at or near the surface of the soil all year or for varying periods of time during the year” and thus, the land is exposed frequently. Screening the data means we include images obtained at low tide, when land emerges from the water. This only allows us to monitor wetlands in areas with sufficient

wetland surface exposure. In this case, monitoring the wetland area, would not be different from any other land cover type. We have provided 2 additional references of InSAR monitoring on peatland and deltas on lines 85 – 86, illustrating the application of InSAR for different landcover types.

Ohenhen et al., have clearly identified an important shortcoming with respect to the lack of VLM in understanding vulnerability to sea-level hazards and developed a novel dataset for addressing it. Their results are suggestive of threats to coastal agriculture and wetlands. However, it would seem prudent to first carefully evaluate the ability of InSAR to extract VLM in coastal wetlands and agricultural lands, through comparisons with in-situ measurements, and to properly quantify the uncertainties of these particularly challenging landcover types in small areas before making large generalizations. That is, to err on the side of caution and attention to detail.

One fundamental limitation in previous wetland assessments, which we seek to address in this study, is the underestimation of subsidence – shallow and deep subsidence. In most studies, only the shallow components are measured and the deep component are accounted for by comparison with RSLR data. This approach is limited and relies on the assumption that the deep component of subsidence is spatially continuous from the few selected tide gages to the wetlands. However, subsidence varies spatially and as included on lines 257 – 258, wetlands show a nonlinear increase in subsidence with accretion. This subsidence is accounted for not in the shallow strata but in the deeper strata. InSAR measurements incorporate shallow and deep subsidence rates, providing the most accurate assessment of wetland elevation change.

Furthermore, our vulnerability assessment is consistent with a recent study by Saintilan, N. et al. (2022), which suggests elevation deficits for some areas on the US Atlantic coast. In addition, we extracted the elevation rate data from Saintilan, N. et al. (2022) for comparison with our InSAR VLM data. We only considered measurements within 10 m radius of our InSAR locations due to spatial variability in the VLM measurement. From the 43 measurements, only 2 SET-MH stations are located within a 10 m radius of at least one wetland pixel. This suggests that the prioritization of SET-MH monitoring stations and the location of InSAR wetland pixels are different. These differences in the wetland pixels and point monitoring stations may perhaps be related to the fact that sites for the SET-MH monitoring stations are influenced by human apriori knowledge and may be located in permanently/frequently inundated zones. While InSAR pixels are based on land surface exposure and favor less inundated areas. Nevertheless, InSAR measurements represent greater subsamples of wetlands on the US Atlantic coast than point measurements. We have included these statements in lines 444 - 459 of the manuscript. The comparison of the elevation rate with InSAR VLM is shown in supplementary Table 3 and shows consistency between SET-MH measurements barring errors due to unaccounted subsidence.

We have tried to address all the reviewer's concerns about the wetland uncertainties and have included a limited comparison to in-situ measurement. We thank the reviewer for their suggestions.

References

Usai, S., et al. (2000), Modelling terrain deformations at the Phlegrean Fields with INSAR, paper presented at IGARSS 2000. IEEE 2000 International Geoscience and Remote Sensing Symposium. Taking the Pulse of the Planet: The Role of Remote Sensing in Managing the Environment. Proceedings (Cat. No. 00CH37120), IEEE.

Meckel, T. A., U. S. ten Brink, & S. J. Williams (2006), Current subsidence rates due to compaction of Holocene sediments in southern Louisiana, *Geophysical Research Letters*, 33(11), doi:10.1029/2006gl026300.

Saintilan, N. et al. (2022), Constraints on the adjustment of tidal marshes to accelerating sea level rise. *Science* 377, 523–527.

Fox-Kemper, B. et al. (2021), Ocean, Cryosphere and Sea Level Change. In: *Climate Change 2021: The Physical Science Basis. Contribution of Working Group I to the Sixth Assessment Report of the Intergovernmental Panel on Climate Change.* (Cambridge Univ. Press, 2021). <https://www.ipcc.ch/report/ar6/wg1/>

Reviewer #3 (Remarks to the Author): (Reviewers comments in normal text, reponse to reviewers are in bold)

I am pleased to see the authors agreed with the suggested changes, and believe the manuscript is stronger for it. I greatly appreciate that a clear description of the derivation and use of all error terms now appears in the methods section, and that the final paragraph of the discussion now emphasises why measuring vertical land motion in large river deltas is important.

I believe the authors have fairly reviewed all suggestions, and justified all amendments appropriately. I have no further comments and would be happy to see this manuscript published.

Your comments were very helpful in improving the manuscript. We thank you again for your comments and excellent suggestions.

REVIEWER COMMENTS

Reviewer #1 (Remarks to the Author):

- I greatly appreciate the effort the authors have made to address my comments. The following remain outstanding.

- The first line of the abstract, "The vulnerability of coastal environments to sea-level rise varies spatially due to local land subsidence" is misleading. Local land subsidence is one component of the spatial variability of relative(!) sea-level rise. Large scale subsidence is the dominant feature (Piecuch et al. 2018, <https://doi.org/10.1038/s41586-018-0787-6>), and ocean/atmospheric properties are also important (Sallenger Jr. et al., 2012, <https://doi.org/10.1038/nclimate1597>; Piecuch & Ponte, 2015, <https://doi.org/10.1002/2015GL064580>).

"What this approach lacks in resolution is accounted for in accuracy."

- In this case, resolution is part of accuracy. I don't see analysis justifying that the impact of the shallow subsidence is greater than the impact of using a correction from far away. Also, it's unclear whether the GPS and tide gauge / tide gauge benchmark are anchored at similar depths.

- Thanks for clearing up the semi/continuous. The vector dataset was in a previous response to me.

IPCC VLM already incorporates projections for VLM from tide gages, which are not contemporary rates of VLM. Here, we project contemporary VLM rates from 2007 – 2020 for the US Atlantic coast which can be incorporated into current RSLR projections for use in inundation models.

- Again, inundation modeling is not part of this study. 'Contemporary' is process-dependent. Given that many processes impacting coastal VLM are anthropogenic - nonlinear - it would seem to make sense to want a longer time-series to minimize spurious signals. The edited sentence now says the averaging in a radius around the gauge is 'to reduce localized high VLM rates', which contradicts the response: 'Varying the radius would have a negligible impact!' This additional uncertainty remains unquantified.

~25% of the temporal period is unsampled by InSAR. This uncertainty isn't quantified.

"At a lower bound of "2 SIGMA ERROR" wetlands show a 54% vulnerability" / "the interpretation is this"

- VLM +/- 2 sigma for wetlands is not significantly different than 0. As such, I interpret it as an arbitrary number +/- RSL that is not meaningful.

"As seen, the standard deviations are too small. This is a well-known issue with InSAR and GPS time series analysis codes that overestimate the precision of deformation fields."

- I don't know what well known issue is being referred to. I agree that conventional methods of uncertainty analysis (e.g., bootstrapping a velocity model) overestimate precision and don't account well for unwrapping and atmospheric errors. However, the value should be based on the LOS time series. The authors have indicated several times that WabInSAR provides this but have not shown it. Results from another Sentinel-1 analysis suggests uncertainties 25% higher than shown here (Cf. Fig 2. Fig 4 in Tay et al., 2022, <https://doi.org/10.1038/s41893-022-00947-z>), Although different [post]/processing strategies are used, Tay et al., 2022 provides raw time series as a further diagnostic.

"Note that the rate of VLM over wetlands changes very gradually over time and is not affected by anthropogenic processes such as groundwater pumping since groundwater is rarely pumped from aquifers underneath wetlands and the primary driver of elevation change is the addition of new sediments and compaction of sediments under their weight."

- Addition of new sediments is affected by anthropogenic processes (Kirwan & Megonigal, 2013, <https://doi.org/10.1038/nature12856>)

- Thanks for including the details on GNSS selection as they are important.
- Figure 2 compares GNSS used during inversion so doesn't resolve the coastal bias in the non-tie-in set.
- In the text, it says rates were obtained from UNR. I assumed these rates and uncertainties were produced with the MIDAS algorithm but now realize this is not stated (ref 93 may refer to the GPS time-series or MIDAS velocities). And now (above Fig 2 in the rebuttal) the authors state GPS velocities were computed from time-series. Quantifying uncertainty from GPS time series is a non-trivial (Blewitt et al., 2016, <https://doi.org/10.1002/2015JB012552>); Santamaría-Gómez and Ray, 2021, <https://doi.org/10.1029/2020JB019541>), and details are unreported.

- "The spatial variability in the rate at which sea level is rising in open sea or further inland are likely to be minimal within similar locations."
While open ocean absolute sea level is coherent on large spatial scales, it can vary due to a dozen or so processes away from the open ocean (Woodworth et al.,). It is particularly affected by vegetation in wetlands, as correctly noted on L. 278-280.

- Screening is unquantified, and there remains the challenge of differential tides. Low tide at the tide gauge (presumably where the screening decision occurs) is not necessarily low tide in the SAR scene. Or conversely, (assuming a SAR scene is tossed when the tide gauge is at low tide), the tide gauge may not be at low tide (so the SAR scene is kept) but the wetland water level may be higher.

- Within the SAR scene there are likely different tidal heights at different times. Water level height and vegetation type are as important if not more than the difference in dielectric content as illustrated in Fig 2 of Lee et al., 2020, <https://doi.org/10.1109/MGRS.2019.2958653>

- I appreciate the additional comparison with the Saintilan results. The RSET measurements in that study only measure shallow subsidence. "However, subsidence varies spatially and ... this subsidence is accounted for not in the shallow strata but in the deeper strata." Is there a reference for this statement from the rebuttal? It would seem to disagree with the broad agreement in the results here and the Saintilan result. The difference in methodology for comparison of VLM to RSET (10 m radius) is inconsistent with the VLM to TG Projection (50 m radius) and unexplained.

Reviewer #1 (Remarks to the Author): (Reviewers comments in normal text, response to reviewers are in bold)

- I greatly appreciate the effort the authors have made to address my comments. The following remain outstanding.

We thank this reviewer for their comments and suggestions. We have addressed the comments as discussed below.

- The first line of the abstract, "The vulnerability of coastal environments to sea-level rise varies spatially due to local land subsidence," is misleading. Local land subsidence is one component of the spatial variability of relative(!) sea-level rise. Large-scale subsidence is the dominant feature (Pieuch et al. 2018, <https://doi.org/10.1038/s41586-018-0787-6>), and ocean/atmospheric properties are also important (Sallenger Jr. et al., 2012, <https://doi.org/10.1038/nclimate1597>; Pieuch & Ponte, 2015, <https://doi.org/10.1002/2015GL064580>).

Our sentence says “the VULNERABILITY of coastal environments varies spatially due to local land subsidence”, we did not refer to RELATIVE SEA LEVEL RISE. On lines 55 – 58 of the manuscript, we discuss processes affecting RELATIVE SEA LEVEL RISE.

There is a consensus that within the next few decades, land subsidence will be the primary driver of relative sea level rise and, thus, flooding hazards (Nicholls et al. 2021 science).

Revised to “The vulnerability of coastal environments to sea-level rise varies spatially, *particularly* due to local land subsidence” in the manuscript on lines 16 and 17.

"What this approach lacks in resolution is accounted for in accuracy."

- In this case, resolution is part of accuracy. I don't see analysis justifying that the impact of the shallow subsidence is greater than the impact of using a correction from far away. Also, it's unclear whether the GPS and tide gauge / tide gauge benchmark are anchored at similar depths.

This comment is misleading and misrepresents two crucial yet distinct parameters, accuracy and resolution. Resolution and accuracy are two completely different elements. Accuracy refers to how close a reported measurement is to the true value being measured, while the resolution is the smallest change that can be measured. Again we would like to reiterate that correction of the tide gauge using InSAR would introduce greater uncertainties by removing vertical land motion (VLM) rates that are not incorporated in the tide gauge measurements. Tide gauge and GPS bench monuments are often built identically so that tide gauge stations can be integrated into the IGS network (for more discussions, see this NOAA report), so they likely measure similar VLM components.

To further illustrate our point, consider the equations below:

$$VR^* = InSAR_VLM_{S,D} - (RSLR - InSAR_VLM_{S,D}) \quad (1)$$

$$VR = InSAR_VLM_{S,D} - (RSLR - GNSS_VLM_D) \quad (2)$$

Where VR is the vertical resilience, VLM is the vertical land motion rates, the subscripts S and D are shallow and deep subsidence in the VLM, and RSLR is the relative sea level rise rates, including sea level rise and deep VLM rates.

From equation (1), if we correct RSLR using the InSAR VLM (containing shallow and deep subsidence), we would simply be representing the VR as the relative SLR, excluding shallow VLM, which is incorrect. Contrarily, we increase the accuracy of the VR measurements by correcting relative SLR using deep subsidence measured by GNSS stations, as shown in equation (2).

- Thanks for clearing up the semi/continuous. The vector dataset was in a previous response to me.

We are glad to clear up this point.

“IPCC VLM already incorporates projections for VLM from tide gages, which are not contemporary rates of VLM. Here, we project contemporary VLM rates from 2007 – 2020 for the US Atlantic coast which can be incorporated into current RSLR projections for use in inundation models.”

- Again, inundation modeling is not part of this study. 'Contemporary' is process-dependent. Given that many processes impacting coastal VLM are anthropogenic - nonlinear - it would seem to make sense to want a longer time-series to minimize spurious signals. The edited sentence now says the averaging in a radius around the gauge is 'to reduce localized high VLM rates', which contradicts the response: 'Varying the radius would have a negligible impact!' This additional uncertainty remains unquantified. ~25% of the temporal period is unsampled by InSAR. This uncertainty isn't quantified.

This comment misrepresents our work. Here, we did not perform inundation modeling but compared rates of VLM used in the IPCC projection with that we obtained here. This is extremely useful to highlight the “hidden vulnerability of US Atlantic coast...” As stated by the reviewer, many of the processes impacting coastal VLM are anthropogenic, but VLM measurements at tide gauges used in the current IPCC projections do not contain these anthropogenic processes.

Here, we chose a 200m radius to perform averaging for reducing the impact of localized high VLM rates due partly to measurement noise' that may arise, which is a widely used practice and varying the radius to incorporate multiple InSAR points would have a negligible impact on the rates, this has been tested using a radius of 200 m to 500 m.

To be consistent with IPCC report, We used a linear projection for VLM and quantified the uncertainty of our projected VLM measurement at the IPCC station by including a likely range for the projected rates using 2 SD, as shown in figure 3 and captions and equation 8. The gap between ALOS and Sentinel period is unavoidable and can not be filled with publically available measurements, so any attempt to assign an uncertainty for the 25% period is guesswork and scientifically unjustified. Nevertheless, we already quantified the uncertainties in the period of the ALOS satellites by using a larger uncertainty for the ALOS observation, as stated on lines 351 to 352 of the manuscript.

"At a lower bound of “2 SIGMA ERROR” wetlands show a 54% vulnerability" / "the interpretation is this"

- VLM +/- 2 sigma for wetlands is not significantly different than 0. As such, I interpret it as an arbitrary number +/- RSL that is not meaningful.

This comment is misleading and contradicts the very foundation of statistics and the theory of errors. The difference between 54% and 100% is meaningful! In this study, we analyzed 1,276 km² of wetland area, at 54% vulnerability, approximating an additional area of 587 km². For comparison, Miami metropolitan has a total area of 145.23 km² (<https://en.wikipedia.org/wiki/Miami>). Considering the value of wetlands as discussed in lines 41 – 48, the loss of additional wetlands within +/-2 sigma is significant and meaningful.

"As seen, the standard deviations are too small. This is a well-known issue with InSAR and GPS time series analysis codes that overestimate the precision of deformation fields."

- I don't know what well known issue is being referred to. I agree that conventional methods of uncertainty analysis (e.g., bootstrapping a velocity model) overestimate precision and don't account well for unwrapping and atmospheric errors. However, the value should be based on the LOS time series. The authors have indicated several times that WabInSAR provides this but have not shown it. Results from another Sentinel-1 analysis suggests uncertainties 25% higher than shown here (Cf. Fig 2. Fig 4 in Tay et al., 2022, <https://doi.org/10.1038/s41893-022-00947-z>), Although different [post]/processing strategies are used, Tay et al., 2022 provides raw time series as a further diagnostic.

The issue we refer to is discussed in the literature as the effect of a “large degree of freedom on mean value standard deviation”. To clarify, consider measuring a length with a ruler with a spacing of 1 mm. The precision of measurements is thus 1 mm. Now, repeat the exercise 10000 times. The standard deviation of the mean reduces to $1\text{mm}/10000^{0.5}$, but no one believes that using a ruler, a length was measured at 10 microns precision. This is the issue in GNSS and INSAR processing when there are many observations and few unknowns, which results in a significantly large degree of freedom and underestimation of errors.

Comparing our work with that of Tay et al., 2022 is irrelevant as the study areas, and processing methods are different. However, we note that the validation of Tay et al. results against GNSS observation (e.g., Fig 4d) yields a very poor agreement. In contrast, here, we yield mm-level accuracy for our results as reported in the man text. Thus, indicating our approach's superiority and our results' reliability.

Our methods and results have been thoroughly validated. We draw attention to our recent paper published in the Journal Remote Sensing of Environment, the flagship of the remote sensing community (Lee & Shirzaei, 2023). In this paper, we carefully validated our processing approach in various settings and yielded remarkable agreement with independent observations in terms of LOS time series and rates. We believe our analysis and results and robust.

"Note that the rate of VLM over wetlands changes very gradually over time and is not affected by anthropogenic processes such as groundwater pumping since groundwater is rarely pumped from aquifers underneath wetlands and the primary driver of elevation change is the addition of new sediments and compaction of sediments under their weight."

- Addition of new sediments is affected by anthropogenic processes (Kirwan & Megonigal, 2013, <https://doi.org/10.1038/nature12856>)

Acknowledged! The addition of new sediments is affected by anthropogenic processes such as dams and reservoirs and the construction of buildings along channels.

- Thanks for including the details on GNSS selection as they are important.

We are glad to clear up this point.

- Figure 2 compares GNSS used during inversion so doesn't resolve the coastal bias in the non-tie-in set.

The reviewer is misinterpreting the figure. As clearly explained in the caption and text, Figure 2 in the manuscript does not compare GNSS used during the inversion, but independent GNSS stations (i.e., validation data set) are shown as stated on lines 389 to 393 of the manuscript. Figure 2 in the rebuttal uses stations from the tie stations and is provided to highlight that the GNSS selection is randomized.

- In the text, it says rates were obtained from UNR. I assumed these rates and uncertainties were produced with the MIDAS algorithm but now realize this is not stated (ref 93 may refer to the GPS time-series or MIDAS velocities). And now (above Fig 2 in the rebuttal) the authors state GPS velocities were computed from time-series. Quantifying uncertainty from GPS time series is a non-trivial (Blewitt et al., 2016, <https://doi.org/10.1002/2015JB012552>); Santamaría-Gómez and Ray, 2021, <https://doi.org/10.1029/2020JB019541>), and details are unreported.

The reviewer misrepresents the MIDAS approach. MIDAS is an approach for automatically estimating velocities and its uncertainties from GPS coordinate time series given the nonlinearities due to seasonal and interannual variations and steps. The exact statement in the rebuttal is: "To estimate the GNSS rate, we considered the nonlinearity in the GNSS dataset by considering semi-annual, annual, and bi-annual terms in addition to the long-term rates." This is embedded in the MIDAS approach (See Blewitt et al. 2018). As stated on lines 353 to 353, we utilized the rates and error measurements from the Nevada Geodetic laboratory; hence, we do not report details for an analysis the authors did not perform.

- "The spatial variability in the rate at which sea level is rising in open sea or further inland are likely to be minimal within similar locations."

While open ocean absolute sea level is coherent on large spatial scales, it can vary due to a dozen or so processes away from the open ocean (Woodworth et al.,). It is particularly affected by vegetation in wetlands, as correctly noted on L. 278-280.

We stated that variability is minimal, not absent. Additionally, there are more significant uncertainties in utilizing RSLR for analysis, mainly due to VLM. This use of RSLR at tide gauge stations is the currently applied methodology for estimating wetland vulnerability. However, we significantly reduce the uncertainty in wetland vulnerability estimates by using absolute SLR, which do contain uncertainties due to VLM invariably embedded in the RSLR.

- Screening is unquantified, and there remains the challenge of differential tides. Low tide at the tide gauge (presumably where the screening decision occurs) is not necessarily low tide in the

SAR scene. Or conversely, (assuming a SAR scene is tossed when the tide gauge is at low tide), the tide gauge may not be at low tide (so the SAR scene is kept) but the wetland water level may be higher. -Within the SAR scene there are likely different tidal heights at different times. Water level height and vegetation type are as important if not more than the difference in dielectric content as illustrated in Fig 2 of Lee et al., 2020, <https://doi.org/10.1109/MGRS.2019.2958653>

This comment is misleading and misrepresents our work. We screened each SAR scene using a statistical framework for flood mapping developed in Sherpa & Shirzaei (2021) to obtain only images with the maximum exposed surface and least water bodies. We have included this on line 296 of the manuscript. As stated in the earlier revisions, wet soils return signals, but flooded wetlands yield no return signals, we did not obtain signals over flooded wetlands.

- I appreciate the additional comparison with the Saintilan results. The RSET measurements in that study only measure shallow subsidence. "However, subsidence varies spatially and ... this subsidence is accounted for not in the shallow strata but in the deeper strata." Is there a reference for this statement from the rebuttal? It would seem to disagree with the broad agreement in the results here and the Saintilan result. The difference in methodology for comparison of VLM to RSET (10 m radius) is inconsistent with the VLM to TG Projection (50 m radius) and unexplained.

As stated by the reviewer, RSET measurements measure shallow subsidence, which is then subtracted from RSLR, since RSLR contains deep subsidence, it is assumed all subsidence processes (shallow and deep) are incorporated in the evaluation of vertical resilience. The comparison with Saintilan's results is the only comparison possible since it is the only study (from our literature review) that records elevation deficits and not just accretion rates. Estimates of vertical resilience made by comparing the accretion rates with RSLR do not incorporate all processes of subsidence ongoing in wetlands. This has been discussed in-depth throughout the manuscript.

“However, subsidence varies spatially and ... this subsidence is accounted for not in the shallow strata but in the deeper strata.” This highlighted statement refers to not subsidence in general but the nonlinear increase in subsidence with accretion. The full sentence in the rebuttal is: “However, subsidence varies spatially and as included on lines 257 – 258, wetlands show a nonlinear increase in subsidence with accretion. This subsidence is accounted for not in the shallow strata but in the deeper strata.”

The use of a radius of 10 m for comparison of VLM to RSET is based on the radius for wetland database to VLM comparison as noted on lines 447 to 448 of the manuscript. Please note that tide gages were only used for the SLR data and not the wetlands database.

References

Lee, J.-C., & Shirzaei, M. Novel algorithms for pair and pixel selection and atmospheric error correction in multitemporal InSAR. *Remote Sensing of Environment*, 286, 113447 (2023).

Nicholls, R. J. et al. A global analysis of subsidence, relative sea-level change and coastal flood exposure. *Nat. Clim. Chang.* 11, 634 (2021).

Sherpa, S. F., & Shirzaei, M. Country-wide flood exposure analysis using sentinel-1 synthetic aperture radar data: case study of 2019 iran flood. *Journal of Flood Risk Management*, 15, (2021).

REVIEWER COMMENTS

Reviewer #4 (Remarks to the Author):

Summary:

The essence of this paper is to apply advanced InSAR time series processing, combining ALOS and Sentinel-1 ascending track frames with GNSS time series to solve for a coastal map or VLM at the resolution of InSAR and tied to the global reference system through GNSS observations. They use ALOS (active 2007-2011) and Sentinel-1 (active since 2015) to estimate the linear rates. From these maps of VLM they compare their VLM time series projections to 2100 with the IPCC 6th report projections for a number of points along the east coast of the US. Finally a significant amount of their findings focus on the vulnerability (in particular the percentage) of coastal marshes that are subsiding relative to sea level rise.

Review:

The paper's results derive completely from their ability to solve for very small rates (a few mm/year or less, typically) along the entire east coast of the US. And, as is well known, RSLR (relative sea level rise) can be dominated by VLM locally where VLM is significant. From a US-centric perspective the results are important and if extended globally the methods applied could have broader impact. The observation of coastal marsh compaction is important, although I did not find a clear connection to process understanding in terms of the cause of the observed subsidence.

In terms of the VLM and RSLR results from which they derive their findings I have a few comments and criticisms.

1. The most significant criticism I have with this analysis is that there are some very clear velocity field edge effects at frame boundaries for both ALOS and Sentinel-1. These can be seen in Fig 2, 4b, 5, and Supplemental Fig. 1 and Supp. Fig. 2, especially in northern N. Carolina and Chesapeake Bay/Delmarva, NY/NJ and Boston areas, where there are straight edge discontinuities (jumps) in VLM that are due to ramps or offsets in the InSAR time series linear rates. Given the subtle nature (small rates) of the VLM signals and the resulting interpretation of these rates with other observations I feel these foundational InSAR rates need improved analysis to support their results.

2. I'm not quite sure why the authors did not also process descending track Sentinel-1 data since this would improve their ability to separate the vertical motion from the E motion (I understand that ALOS did not acquire significant amounts on descending tracks).

3. Page 5, lines 183-185: I am a little uncertain what to make of these numbers? Is this significant or not? In other words, if most of the land within a certain distance of the ocean along the east coast is wetlands and forest, then it's no surprise that those are the areas subsiding (assuming these are the areas most naturally subject to compaction of sediments, etc.).

4. Lines 211-212, page 6: I can't see any difference between the high and low wetland maps. As noted in the caption the reader is pointed to the zoomed figures in supplementary Fig. 6. However, here too, although differences can be seen upon close scrutiny they are very subtle. As I mentioned above, in Fig 5 a & c there are some sharp linear boundaries that correspond to Sentinel-1 and/or ALOS frame boundaries. These appear to be time series errors that lead to adjacent areas of + and - VR. These are clearly artifacts that should be corrected.

5. Line 306 in SAR Analysis: By definition you really have no topography in your study area! So this is a capability in your methodology that really has no relevance here since your major problems lie with turbulent atmospheric effects (which you diminish through temporal smoothing). One large error source, especially for the L-band ALOS data are ionospheric effects. Do you remove these through split spectrum filtering or other means (throwing out strongly affected scenes)?

6. Line 328: GNSS stations with only 50 measurements per year? I.e. less than 1 sample per week? Seems a particularly low bar for inclusion, and if a GNSS site had only 50 samples in a year that would not seem to be very reliable.

7. Figure 3 caption, line 809, regarding the percent differences in each subplot: Your numbers don't seem correct: first in the equation the division by 2 in the denominator seems like it should be not just $\text{InSARVLM}/2$ but $(\text{IPCCVLM} + \text{InSARVLM})/2$. If I take (e) as an example applying the formula it is $(3.5 -$

$2.0 / ((3.5 + 2.0) / 2) = 55\%$ so I have no idea where you get 13.6%

8. Supplemental Fig. 5 caption, line 141, regarding "The linear rate is 0.99 mm/yr." which references the rate of hydraulic head change in water wells from 2014 to 2020.3: From the plot we see (in subplot (c)) that there is at least 1 m of increase in head level (decrease in the depth to the water level from the surface) which over ~6 years equals ~0.15-0.2 m/yr (150-200 mm/yr), many orders of magnitude than the 1 mm/yr stated in the figure. Please explain.

Minor points/text corrections:

1. The satellite from JAXA is ALOS, which was followed by ALOS-2 in 2014 (with ALOS-4 the follow-on to that). Nit-picky I know, but for those of us processing ALOS data when there was no follow-on mission, it was just ALOS, not ALOS-1.

2. Line 53: should read "...1 m or more by 2100..."

3. Line 96: replace "obtained" with "these"

4. Line 116: delete "of"

5. Line 168: Should read "The areas the NLC noted..."

6. Line 331: "...projections GNSS 3D velocities on Sentinel-1..."

7. Line 460: "...greater number of subsamples..."

8. Line 810: "100." (not 100%, which is just 1)

Reviewer #4 (Remarks to the Author): (Reviewers comments in normal text, response to reviewers are in bold)

Summary:

The essence of this paper is to apply advanced InSAR time series processing, combining ALOS and Sentinel-1 ascending track frames with GNSS time series to solve for a coastal map or VLM at the resolution of InSAR and tied to the global reference system through GNSS observations. They use ALOS (active 2007-2011) and Sentinel-1 (active since 2015) to estimate the linear rates. From these maps of VLM they compare their VLM time series projections to 2100 with the IPCC 6th report projections for a number of points along the east coast of the US. Finally a significant amount of their findings focus on the vulnerability (in particular the percentage) of coastal marshes that are subsiding relative to sea level rise.

Review:

The paper's results derive completely from their ability to solve for very small rates (a few mm/year or less, typically) along the entire east coast of the US. And, as is well known, RSLR (relative sea level rise) can be dominated by VLM locally where VLM is significant. From a US-centric perspective the results are important and if extended globally the methods applied could have broader impact. The observation of coastal marsh compaction is important, although I did not find a clear connection to process understanding in terms of the cause of the observed subsidence.

We appreciate the comments and note that for this paper, we focused on the observed observations and wetlands exposure analysis due to their importance. Due to the extent of the study area and the fact that various natural and anthropogenic processes act in concert to drive the observed VLM, investigating the driving factors requires a separate study, which is a subject of ongoing research in our lab.

In terms of the VLM and RSLR results from which they derive their findings I have a few comments and criticisms.

1. The most significant criticism I have with this analysis is that there are some very clear velocity field edge effects at frame boundaries for both ALOS and Sentinel-1. These can be seen in Fig 2, 4b, 5, and Supplemental Fig. 1 and Supp. Fig. 2, especially in northern N. Carolina and Chesapeake Bay/Delmarva, NY/NJ and Boston areas, where there are straight edge discontinuities (jumps) in VLM that are due to ramps or offsets in the InSAR time series linear rates. Given the subtle nature (small rates) of the VLM signals and the resulting interpretation of these rates with other observations I feel these foundational InSAR rates need improved analysis to support their results.

We thank the reviewer for this observation. The edge effect noted do not arise from the InSAR processing, such as ramp effects or offsets in the InSAR time series linear rates, but are unavoidable edge effects from the mosaic process of the ALOS frames, with a variable number of observations that are noticeable on the ALOS line-of-sight (LOS) velocity (Supplementary Figure 1b). To mitigate the impact of the edge effect on the final results, we considered two remedies; 1) we perform a joint inversion of ALOS, Sentinel-1, and GNSS measurements, in that GNSS observations act as benchmarks and avoid error propagation, 2) we down-weight ALOS LOS velocities, compared with Sentinel-1, in

particular wherever there is an obvious issue with the measurements. The effectiveness of our approach is evident from the final results (Fig 2, S2), which are smooth and do not exhibit edge effects.

Also, note that the discontinuity mentioned by the reviewer is exaggerated in Figures 2b and 4b due to the choice of color bar to only color code subsiding pixels; thus, it is merely a plotting issue. Figure 1 below shows the plot for the Chesapeake Bay area, with all pixels color-coded. Figure 1b and c show zoomed-in areas at the top and the bottom of the map to show continuity in the velocity. Also, noticeable on figure 1c is a gap between adjacent pixels from different ALOS frames. When viewed from a distance, this gap is reflected as a discontinuity in velocity between adjacent pixels. However, as shown in Figure 1c, there are consistencies between the velocities in the adjacent pixel. The sharp boundaries noted in Fig. 5a and c do not arise from the VLM, but are a result of linearly interpolated absolute sea level rates on the InSAR pixels. Furthermore, in our analysis, including the vulnerability estimates, we accounted for all uncertainties in our VLM using a 95% confidence interval.

[redacted]

Figure 1: Vertical land Motion of Chesapeake Bay area.

Note that other observed high velocities noted along the coastlines for the Chesapeake Bay, New York, Boston, and North Carolina are not edge discontinuities but accurate velocity measurements in the study area. The U.S. coastline is bordered by so called barren lands and coastal fronts, the majority of which have been developed. The increase

in sea level has caused a retreat of the barren lands due to rapid erosion along the U.S. east coast, especially in New York, Chesapeake Bay, North Carolina, etc. (Fig. 2). The observed rates in our InSAR measurements reflect this erosion as explained on lines 170 – 171.

[redacted]

Figure 2: Beach erosion along the U.S. east coast. The top image is from Bethany Beach Delaware, the bottom image is from North Carolina. (see recent CNN article here: <https://www.cnn.com/2022/02/09/us/home-collapse-north-carolina-climate/index.html>)

2. I'm not quite sure why the authors did not also process descending track Sentinel-1 data since this would improve their ability to separate the vertical motion from the E motion (I understand that ALOS did not acquire significant amounts on descending tracks).

We thank the reviewer for this observation. We explored the complete suite of sentinel-1 images. However, for the U.S. east coast, there were no descending Sentinel-1 images covering our entire study region.

3. Page 5, lines 183-185: I am a little uncertain what to make of these numbers? Is this significant or not? In otherwards, if most of the land within a certain distance of the ocean along the east coast is wetlands and forest, then it's no surprise that those are the areas subsiding (assuming these are the areas most naturally subject to compaction of sediments, etc.).

These numbers help to highlight land cover exposure on the U.S. east coast. It signifies that the dominant land uses are forested land and wetlands. As suggested, this is not surprising due to natural subsidence processes. Perhaps, the use of overwhelmingly in this sentence suggests an exaggeration of this point. We have modified this sentence and removed 'overwhelmingly'.

4. Lines 211-212, page 6: I can't see any difference between the high and low wetland maps. As noted in the caption the reader is pointed to the zoomed figures in supplementary Fig. 6. However, here too, although differences can be seen upon close scrutiny they are very subtle. As I mentioned above, in Fig 5 a & c there are some sharp linear boundaries that correspond to Sentinel-1 and/or ALOS frame boundaries. These appear to be time series errors that lead to adjacent areas of + and – VR. These are clearly artifacts that should be corrected.

Thank you for this observation. The reviewer has spotlighted one of the key highlights of our study. Previous studies suggest that low- versus high-elevation wetlands are not equally vulnerable to relative sea level rise because marsh accretion and vertical vulnerability can be highly dependent on marsh elevation (see lines 207 – 208 and ref. 64). In this study, we show that when elevation deficits on wetlands are accounted for, the vulnerability of wetlands show little dependence on existing marsh elevation and both low- and high-elevation wetlands may be vulnerable to relative sea level rise (see lines 213 – 214). As stated above, the sharp boundaries in Fig. 5a and c are not the result of the VLM, but arise from the interpolation of the absolute sea level, which are linearly interpolated on the InSAR pixels.

5. Line 306 in SAR Analysis: By definition you really have no topography in your study area! So this is a capability in your methodology that really has no relevance here since your major problems lie with turbulent atmospheric effects (which you diminish through temporal smoothing). One large error source, especially for the L-band ALOS data are ionospheric effects. Do you remove these through split spectrum filtering or other means (throwing out strongly affected scenes)?

Thank you for your observation. Geometrical correction of interferogram using precise satellite ephemeris and existing digital elevation model is a standard procedure to create differential interferogram. Even for a flat area such as the east coast, the effect of topography needs to be considered since the SAR interferometry is performed concerning reference ellipsoid or geoid. However, for pairs with short perpendicular baseline (as is the case for Sentinel), the effect of topography can be negligible. We corrected interferograms for the effects of the atmosphere delay using the techniques described in Lee & Shirzaei (2023) (citation 91) and applied ionospheric correction where needed (Zhang et al., 2022; citation 92).

6. Line 328: GNSS stations with only 50 measurements per year? I.e. less than 1 sample per week? Seems a particularly low bar for inclusion, and if a GNSS site had only 50 samples in a year that would not seem to be very reliable.

For this analysis, we set the criteria to at least 50 measurements per year to increase the number of GNSS stations utilized in the analysis. Most stations (79%) have at least 300 samples per year. In fact, there are no GNSS stations utilized in our analysis with less than 100 samples per year (Figure 3). We have included this plot in supplementary figure 9 and referenced it on line 329 of the manuscript.

Figure 3: Minimum number of samples per year for each GNSS station utilized in our analysis.

7. Figure 3 caption, line 809, regarding the percent differences in each subplot: Your numbers don't seem correct: first in the equation the division by 2 in the denominator seems like it should be not just $\ln(\text{SARVLM}/2)$ but $(\ln(\text{IPCCVLM} + \ln(\text{SARVLM}))/2)$. If I take (e) as an example applying the formula it is $(3.5 - 2.0) / ((3.5 + 2.0)/2) = 55\%$ so I have no idea where you get 13.6%

We thank the reviewer for their keen observation. This was a calculation error and has been corrected in figure 3 and the supplementary table 2.

8. Supplemental Fig. 5 caption, line 141, regarding "The linear rate is 0.99 mm/yr." which references the rate of hydraulic head change in water wells from 2014 to 2020.3: From the plot we see (in subplot (c)) that there is at least 1 m of increase in head level (decrease in the depth to the water level from the surface) which over ~6 years equals ~0.15-0.2 m/yr (150-200 mm/yr), many orders of magnitude than the 1 mm/yr stated in the figure. Please explain.

We thank you immensely for this observation. Our reported linear rate is mm/day not mm/year. We have modified this in supplementary Fig. 5.

Minor points/text corrections:

1. The satellite from JAXA is ALOS, which was followed by ALOS-2 in 2014 (with ALOS-4 the follow-on to that). Nit-picky I know, but for those of us processing ALOS data when there was no follow-on mission, it was just ALOS, not ALOS-1.

We have corrected all occurrences in the main manuscript including the figure 1 to reflect ALOS and not ALOS-1.

2. Line 53: should read "...1 m or more by 2100..."

This sentence has been modified on line 53.

3. Line 96: replace "obtained" with "these"

We replaced "obtained" with "these" on line 96.

4. Line 116: delete "of"

Deleted "of" on line 116.

5. Line 168: Should read "The areas the NLC noted..."

This sentence has been modified on line 168.

6. Line 331: "...projections GNSS 3D velocities on Sentinel-1..."

This sentence refers to the north, east, vertical velocity onto the InSAR line-of sight and not the GNSS data.

7. Line 460: "...greater number of subsamples..."

This sentence has been modified on line 460.

8. Line 810: "100." (not 100%, which is just 1)

This sentence has been modified on line 801.

References

Lee, J.-C., & Shirzaei, M. Novel algorithms for pair and pixel selection and atmospheric error correction in multitemporal InSAR. *Remote Sensing of Environment*, 286, 113447 (2023).

Zhang, B., Zhu, W., Ding, X., Wang, C., Wu, S. & Zhang, Q. A review of methods for mitigating ionospheric artifacts in differential SAR interferometry. *Geodesy and Geodynamics* 13, 160–169 (2022).

REVIEWERS' COMMENTS

Reviewer #4 (Remarks to the Author):

The newly revised manuscript has responded to my review and is now acceptable.

Paul Lundgren

Reviewer #4 (Remarks to the Author): (Reviewers comments in normal text, reponse to reviewers are in bold)

The newly revised manuscript has responded to my review and is now acceptable.

Paul Lundgren

Thank you, Dr. Lundgren! We appreciate your review and comments.